# Repeated inversions within a *pannier* intron drive diversification of intraspecific colour patterns of ladybird beetles

Toshiya Ando [1,2], Takeshi Matsuda[3], Kumiko Goto[3], Kimiko Hara[3], Akinori Ito[3], Junya Hirata[3], Joichiro Yatomi[3], Rei Kajitani [4], Miki Okuno[4], Katsushi Yamaguchi[5], Masaaki Kobayashi[6], Tomoyuki Takano[6], Yohei Minakuchi[7], Masahide Seki[8], Yutaka Suzuki[8], Kentaro Yano[6], Takehiko Itoh[4], Shuji Shigenobu [2,5], Atsushi Toyoda [7,9] & Teruyuki Niimi[1,2,3]

How genetic information is modified to generate phenotypic variation within a species is one of the central questions in evolutionary biology. Here we focus on the striking intraspecific diversity of >200 aposematic elytral (forewing) colour patterns of the multicoloured Asian ladybird beetle, *Harmonia axyridis*, which is regulated by a tightly linked genetic locus *h*. Our loss-of-function analyses, genetic association studies, de novo genome assemblies, and gene expression data reveal that the GATA transcription factor gene *pannier* is the major regulatory gene located at the *h* locus, and suggest that repeated inversions and *cis*-regulatory modifications at *pannier* led to the expansion of colour pattern variation in *H. axyridis*. Moreover, we show that the colour-patterning function of *pannier* is conserved in the seven-spotted ladybird beetle, *Coccinella septempunctata*, suggesting that *H. axyridis'* extraordinary intraspecific variation may have arisen from ancient modifications in conserved elytral colour-patterning mechanisms in ladybird beetles.

[1] Division of Evolutionary Developmental Biology, National Institute for Basic Biology, Okazaki, Aichi 444-8585, Japan. [2] Department of Basic Biology, School of Life Science, SOKENDAI (The Graduate University for Advanced Studies), Okazaki, Aichi 444-8585, Japan. [3] Laboratory of Sericulture and Entomoresources, Graduate School of Bioagricultural Sciences, Nagoya University, Nagoya, Aichi 464-8601, Japan. [4] Department of Biological Information, Tokyo Institute of Technology, Meguro-ku, Tokyo 152-8550, Japan. [5] NIBB Core Research Facilities, National Institute for Basic Biology, Okazaki, Aichi 444-8585, Japan. [6] Bioinformatics Laboratory, Department of Life Sciences, School of Agriculture, Meiji University, Kawasaki, Kanagawa 214-8571, Japan. [7] Comparative Genomics Laboratory, National Institute of Genetics, Mishima, Shizuoka 411-8540, Japan. [8] Laboratory of Systems Genomics, Department of Computational Biology and Medical Sciences, Graduate School of Frontier Sciences, The University of Tokyo, Kashiwa, Chiba 277-8562, Japan. [9] Advanced Genomics Center, National Institute of Genetics, Mishima, Shizuoka 411-8540, Japan. Correspondence and requests for materials should be addressed to T.N. (email: niimi@nibb.ac.jp)

There are approximately 6000 ladybird beetle species described worldwide[1]. Charismatic and popular, ladybird beetles are famous for the red and black spot patterns on their elytra (forewings), thought to be a warning signal to predators that they store bitter alkaloids in their body fluids[2,3] and are unpalatable. This red/black warning signal is shared among many ladybird beetle species, and provides a model for colour pattern mimicry by other insect orders. While most ladybird beetle species have only a single spot pattern, a few display remarkable intraspecific diversities, such as the multicoloured Asian ladybird beetle, *Harmonia axyridis*, which exhibits >200 different elytral colour forms (Fig. 1a). This striking intraspecific variation prompted us to investigate its genetic and evolutionary basis.

The first predictions regarding the genetics underlying the highly diverse elytral colour patterns of *H. axyridis* and its relevance to speciation were made by the evolutionary biologist, Theodosius Dobzhansky based on his comprehensive classification of specimens collected from various regions in Asia[4]. Successive genetic analyses[5–7] revealed that many of these colour patterns are actually regulated by a tightly linked genetic locus, *h*, which segregates either as a single gene, or as strongly linked pseudoallelic genes (a supergene[8,9]) (Fig. 1b, c). The elytral colour patterns are assumed to be formed by the superposition of combinations of two of the four major allelic patterns and dozens of minor allelic colour patterns (>20 different allelic patterns in total). The major allelic patterns cover more than 95% of colour patterns in the natural population[4]. In the elytral regions where the different colour elements are overlapped in heterozygotes, black colour elements are invariably dominant against red colour elements (mosaic dominance[10]). Whether all of the supposed alleles linked to the *h* locus correspond to a single gene or multiple genes is unknown. Elucidating the DNA structure and the mechanisms underlying the evolution of this tightly linked genetic locus that encodes such a strikingly diverse intraspecific colour pattern polymorphism would provide a case-study that bears upon a major evolutionary developmental biology question; how does morphology evolve?

Here we show that the gene *pannier* is responsible for controlling the major four elytral colour patterns of *H. axyridis*. Moreover, we illustrate how modification to this ancient colour-patterning gene likely contributed to an explosive diversification of colour forms.

## Results

**Elytral pigmentation during *H. axyridis* pupal development.** To identify the gene regulating elytral colour pattern formation of *H. axyridis*, we first investigated the pigmentation processes during development. In the developing pupal elytra, red pigment (carotenoids[11]) was accumulated in the future red-pigmented regions (Fig. 2a, pharate adult elytron). Red pigmentation occurred only in the thick ventral epidermal cells of the two layers of the elytral epidermis (Fig. 2b, c, red), and started at 80 h after pupation (80 h AP). Black pigmentation (melanin accumulation[11]) occurred only in the dorsal cuticle of black-pigmented regions (Fig. 2d, black), and started approximately 2 h after eclosion. Although pharate adult elytra are not black, we detected a strong upregulation of enzymatic activity related to melanin synthesis[12] in the nascent dorsal cuticle in the future black regions from 80 h AP (Fig. 2a, lower panels; Fig. 2c, black; Supplementary Figure 1). Every black-pigmented region was deployed complementary to the red regions. Therefore, we concluded that the developmental programs for both red and black pigmentation started around 80 h AP.

***pannier* promotes melanin and represses carotenoids in elytra.** We hypothesised that some of the conserved genes essential for insect wing/body wall patterning[13–18] are recruited to regulate these elytral pigmentation processes, and tested this possibility using larval RNAi[19]. We performed a small-scale candidate screening focusing on genes involved in wing/body wall patterning (Supplementary Table 1), and found that the *Harmonia* orthologue of *Drosophila pannier*, which encodes a GATA transcription factor[20], is essential for formation of all of the black-pigmented regions in the elytra. For all four major *h* allele

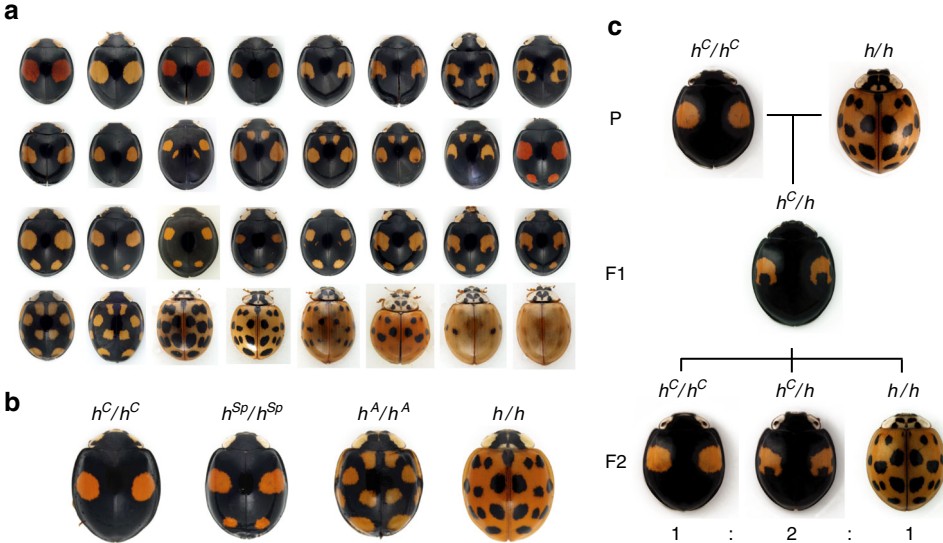

**Fig. 1** Intraspecific genetic polymorphisms of elytral colour patterns in *H. axyridis*. **a** Highly diverse elytral colour patterns of *H. ayridis*. **b** Four major alleles of the elytral colour patterns. $h^C$, conspicua; $h^{Sp}$, spectabilis; $h^A$, axyridis; *h*, succinea. **c** An example of inheritance of elytral colour forms. When $h^C/h^C$ and *h/h* are crossed (P), all F1 progenies show the colour pattern of $h^C/h$. Note the small black spots within the red spots in the F1 progeny. When the F1 heterozygotes are sibcrossed, F2 progeny shows three phenotypes ($h^C/h^C$, $h^C/h$ and *h/h*) at the 1:2:1 ratio predicted for Mendelian segregation of a single locus. Inheritance of any combination of colour patterns follows this segregation pattern. Images in Fig. 1a are used under license from Insect DNA Research Society Japan (newsletter, vol. 13, September 2010). All rights reserved

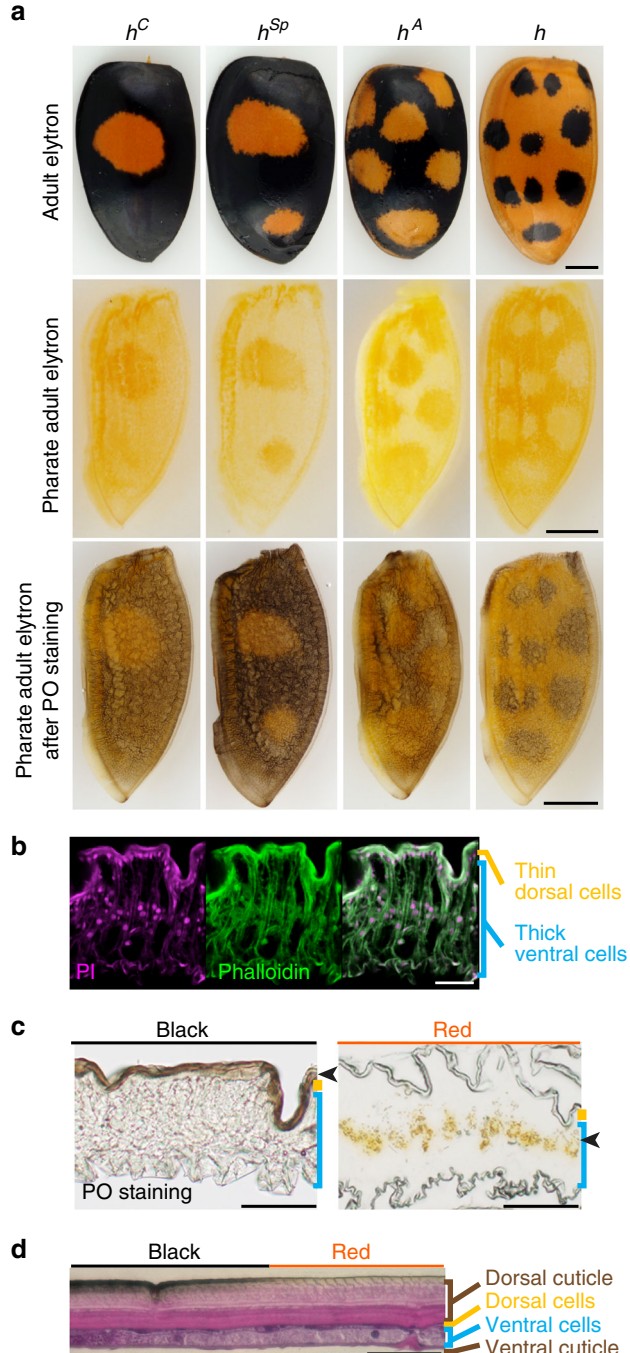

**Fig. 2** Developmental programs for elytral pigmentation initiate at the late pupal stage. **a** Adult elytral colour patterns (upper panels), and localisation of carotenoid (middle panels, orange) and phenol oxidase (PO) activity (lower panels, black) in pharate adult elytra at 96 h AP. Proximal is up. Outer rims are to the left. **b–d** Cross sections of elytra at 96 h AP (**b**, **c**) and adult (**d**). Dorsal is up. **b** magenta, nuclei (propidium iodide); green, F-actin (phalloidin). **c** left, PO staining; right, No staining. Arrowheads indicate pigmented areas. **d** Haematoxylin and eosin staining. Scale bars, 1 mm in (**a**), 50 μm in (**b–d**)

backgrounds, larval RNAi targeting *pannier* resulted in complete loss of black colour elements and alternative emergence of red colour elements in the elytra (Fig. 3a, *H. axyridis*), indicating that *pannier* is essential for inducing black pigmentation in dorsal elytral cells and suppressing red pigmentation in ventral elytral cells. This result was unexpected because *pannier* is not essential

for wing blade patterning in *Drosophila*, but rather essential for patterning of the dorsal body plate attached to the wings (notum) [21–23]. *pannier* mRNA was upregulated from 48 h AP to 96 h AP in elytra (Supplementary Figure 2a), and preferentially in black regions ($h^C$, Supplementary Figure 2b, b′). Immediately before or after 80 h AP (start of the pigmentation program, 76–84 h AP), *pannier* seemingly showed higher expression in the future black regions in the dorsal elytral epidermis (Fig. 3b). These data suggest that region-specific upregulation of *pannier* during the pupal stage regulates black pigmentation in the ladybird beetle's dorsal elytral cells, and that regions of expression differ among the major *h* alleles to form different black patterns in *H. axyridis*.

**The genomic basis for the colour pattern polymorphism**. These data led us to test whether *pannier* is associated with the classically identified locus *h*, which regulates elytral colour patterns. To identify DNA sequences near the *h* locus, we assembled de novo genome sequences (assembly version 1: 423 Mb; contig N50, 63.5 kb; scaffold N50, 1.6 Mb), and performed a genetic association study using the strains with different *h* alleles. We obtained the scaffold containing *pannier* and two additional adjacent scaffolds based on the truncated gene structures at the scaffold ends (Fig. 4a, *Bgb* and *pnr*). Restriction-site Associated DNA Sequencing (RAD-seq) analysis of backcrossed progenies (BC1, $h^A \times h^C$ F0 cross, $n = 183$) revealed that these three scaffolds are included in the five scaffolds that showed complete association with colour patterns (Fig. 4a, the upper left panel). In addition, genotyping of F2 individuals from two other independent genetic sib-crosses ($h^C \times h$ F0 cross ($n = 80$) and $h^A \times h^{Sp}$ F0 cross ($n = 273$)) indicated that the *pannier* locus is included in the relevant regions of all of the major four *h* alleles ($h^C$, 690 kb; $h^{Sp}$, 750 kb; $h^A$, 660 kb; *h*, >2.1 Mb) (Fig. 4a, Supplementary Data 1).

To test contiguity of these three scaffolds, we re-assembled the genome using a novel genome assembler (Platanus2), and performed additional de novo genomic assemblies of $h^C$, $h^A$ and *h* alleles using linked-read and long read sequencing platforms (10× Genomics Chromium system; PacBio system). We obtained contiguous longer genomic scaffolds including the three described above ($h^C$, 3.13 Mb/2.74 Mb; $h^A$, 1.42 + 1.61 Mb; *h*, 2.79 Mb) (Supplementary Figure 3; Supplementary Data 2, *H. axyridis*) and the genotyping markers showing complete association with colour patterns and incomplete association at both ends ($h^C$ and *h*) or one end ($h^A$) of each scaffold (Supplementary Figure 4a–c). These data support the result of our genetic association studies.

To further delimit the candidate genes associated with the elytral colour patterns, we performed RNA-seq analysis using epidermal tissues isolated from the developing red or black regions before pigmentation in the $h^C$ genetic background (Fig. 4b, 24 and 72 h AP RNA-seq). We found that *pannier* was the only gene statistically significantly upregulated in the developing black region compared to the red region at 72 h AP within the *h* locus candidate region (Fig. 4b, red bars, false discovery rate (FDR) < 0.01; Supplementary Data 3). These data pinpoint *pannier* as the major gene regulating the elytral colour pattern variation in *H. axyridis*.

**Inversions and high diversification within a *pannier* intron**. We next investigated allele-specific polymorphisms at the *pannier* locus. We found that alleles of the first intronic region of *pannier* are more diverse than the surrounding genomic regions (Fig. 5a, asterisk, the middle whitish regions in $h^C$ vs. $h^A$, $h^A$ vs. *h*, and *h* vs. $h^C$ comparisons), whereas the same allele in different strains shows conserved fragments distributed throughout the region (Fig. 5a, blue bars, $h^C$ (F2-3) vs. $h^C$ (NT6) comparison). In

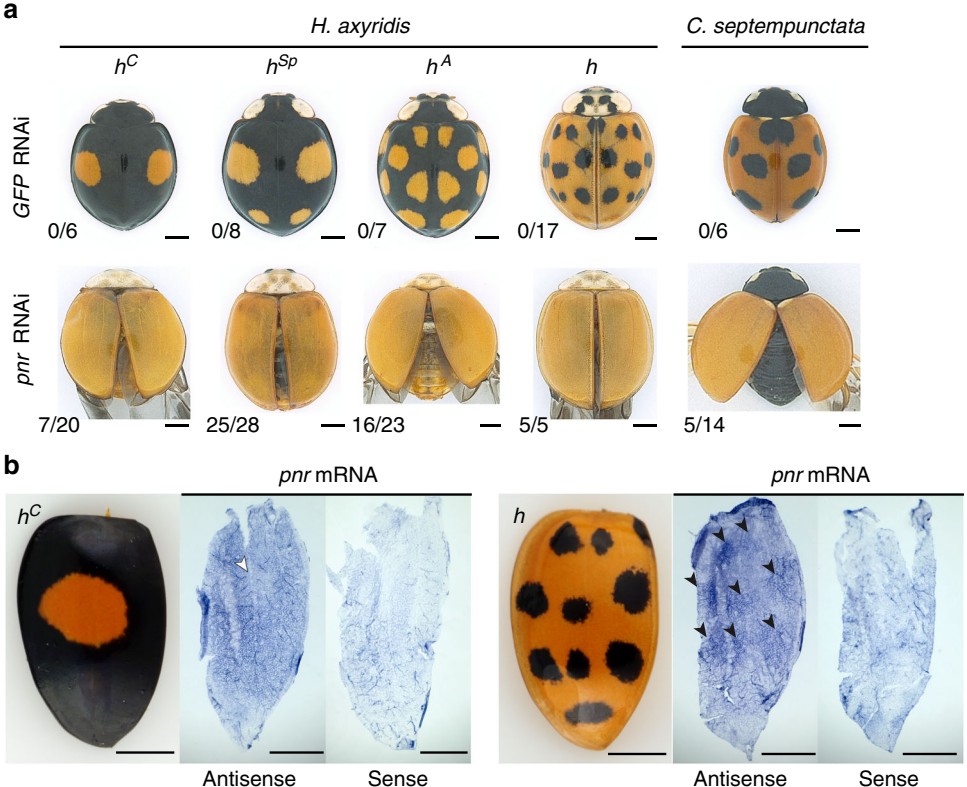

**Fig. 3** *pannier* expression foreshadowing the adult colour pattern switches red/black pigmentation processes. **a** The adult phenotypes of RNAi treatments targeting *GFP* (negative controls, *GFP* RNAi) and *pannier* (*pnr* RNAi) in *H. axyridis* ($h^C$, $h^{Sp}$, $h^A$, $h$) and *C. septempunctata*. Scores in the lower left corners indicate penetrance of the loss-of-pattern phenotype in surviving animals. **b** The pattern of *pannier* expression (*pnr*) in the dorsal elytral epidermal cells immediately before or after pigmentation (76−84 h AP). Left panels indicate the corresponding adult elytral phenotypes ($h^C$ and $h$) adapted from Fig. 2a. White arrowhead, the region with a weak signal. Black arrowheads, the regions with intense signals. Scale bars, 1 mm

comparisons between the alleles, we consistently found traces of large inversions in the upstream half of the first intron (Fig. 5a, reddish lines, $h^C$ vs. $h^A$, $h^A$ vs. $h$, $h$ vs. $h^C$; 56 kb–76 kb in size) (Supplementary Figure 5, dot-plot). However, we found that the coding sequences of *pannier* only showed a single nonsynonymous substitution in the region not conserved among organisms (G235V, $h^{Sp}$) (Supplementary Figure 6, 7), suggesting that *cis*-regulatory differences in the first intronic region of *pannier* are the major cause of intraspecific colour variation.

Moreover, we found that in *H. axyridis*, the size of the upstream noncoding sequences of the *pannier* locus (including the first intron of *pannier*, and the upstream intergenic region between the 5′ end of *pannier* 5′ UTR and 3′ end of the *GATAe* 3′ UTR) are 46–65 kb larger than the currently available corresponding genomic sequences of the other holometabolous insects (Fig. 5b, *H. axyridis*, 153–172 kb; other holometabolous insects, 13−107 kb). Comparison of the exon−intron structures of *H. axyridis* to those of some of the holometabolous insects also suggested that especially the first intron of *pannier* is expanded in *H. axyridis* (*H. axyridis*, 108–118 kb; the other holometabolous insects, 11–44 kb). The expanded region in *H. axyridis* included at least four transcription initiation sites of *pannier* transcripts (Fig. 4c, *pnr-1A–4B*). In addition, in this region, several known DNA-binding motifs of transcription factors involved in *Drosophila* wing formation were more enriched allele-specifically than those in the other genomic regions (Table 1, allele-specifically enriched motifs; Supplementary Data 4). For example, the highly conserved Scalloped (SD) DNA-binding motif of the insect wing selector transcription factor complex Vestigial/Scalloped[24,25] occurred frequently in the upstream and the downstream regions of the first intron of *pannier* specifically

in the $h^C$ allele (Table 1, allele-specifically enriched motifs, $h^C$, Sd in the upstream and the downstream regions of the first intron). Furthermore, the RNA-seq data for the $h^C$ background also revealed that the *sd* coactivator gene *vestigial* was the only transcription factor gene that was significantly upregulated in the future black region from early pupal stages (Supplementary Figure 8), implicating Vestigial as one of the upstream *trans*-regulatory factors acting together with Sd to form the two-spotted elytral colour pattern of $h^C$. It is noteworthy that the noncoding region of *pannier* in each allele possesses putative DNA-binding motifs that can respond to variety of developmental contexts such as anterior−posterior patterning[26–28] (En, Inv), wing fate specification[17,24,29] (Sd), hinge-wing blade patterning[17,18,30,31] and wing vein patterning[29,32–34] (Ab, Al, B-H1, B-H2, Brk, Exd, H, Hth, Kni, Mad, Med, Nub, Rn, Ss, Vvl), hormonal cues[35,36] (EcR, Tai, Usp), and auto-regulation (Pnr) (Table 1, allele-specifically enriched motifs). These results suggest that allele-specific elytral colour patterns of *H. axyridis* may be formed by integrating appropriate combinations of developmental contexts of wing formation shared among insects.

**Colour-patterning function of *pannier* conserved in ladybirds.**
We further tested whether the regulatory function of the red/black colour pattern in elytra is a conserved or a derived aspect of *pannier* function in ladybird beetles using the seven-spotted ladybird beetle, *Coccinella septempunctata*, which shows a monomorphic seven-spotted elytral colour pattern. The *pannier* mRNA was detected in the larval elytral primordium, was upregulated from 24 h AP to 96 h AP (Supplementary Figure 9a), and preferentially expressed in the black spots of elytra in *C. septempunctata* (Supplementary Figure 9b, b′) similarly to that in *H.*

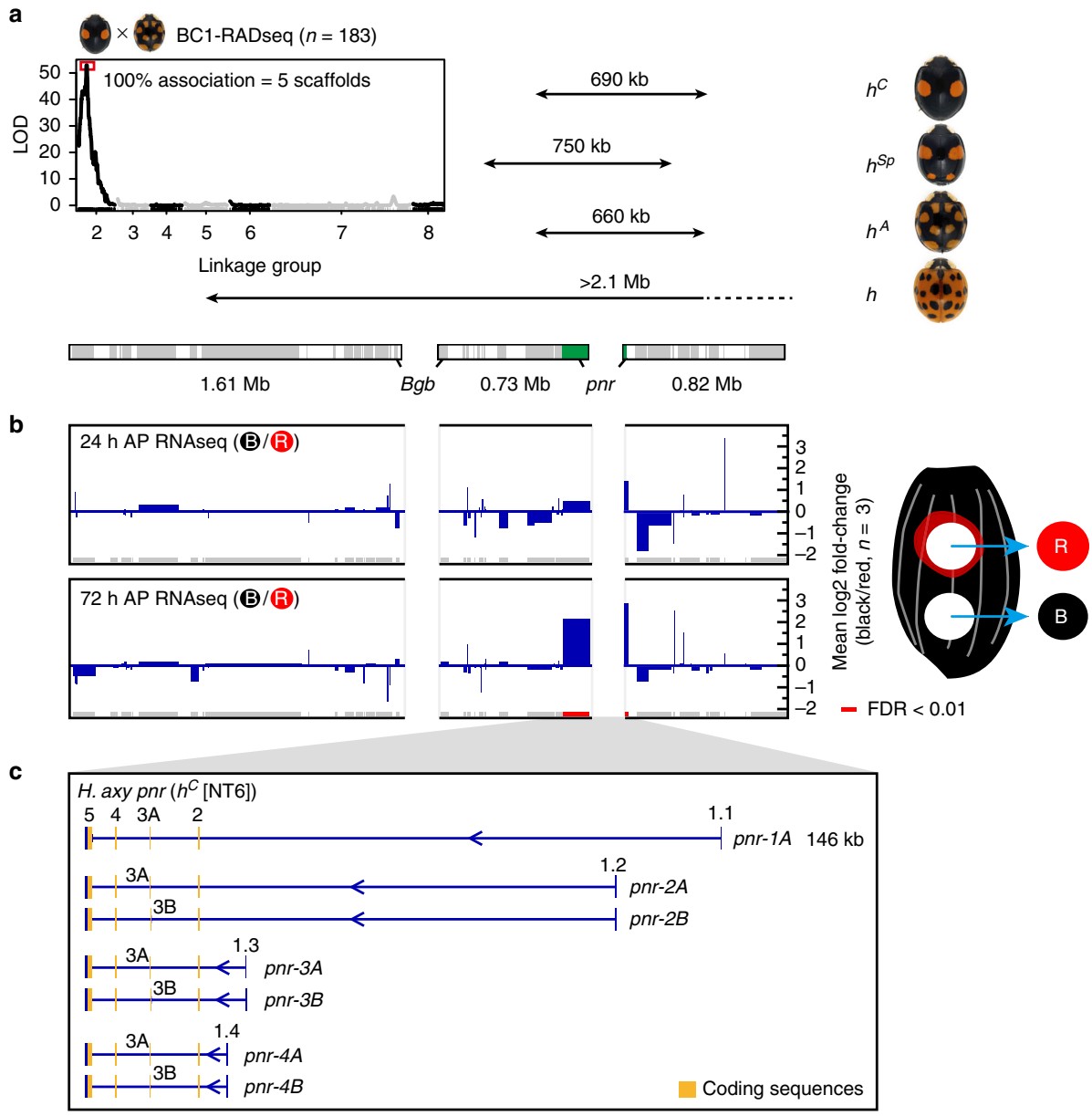

**Fig. 4** *pannier* is the major elytral colour pattering gene located at the *h* locus. **a** Genetic association studies of the *h* locus. Upper left panel, LOD plot for 4419 RAD tags deduced from genotyping BC1 progeny from the $h^C$−$h^A$ F0 cross. RAD tags showing segregation patterns of markers located on the X chromosome were excluded from the analysis. The LOD score peaked at RAD tags in Linkage Group 2. RAD tags with complete association with elytral colour patterns corresponded to five genomic scaffolds including the three scaffolds adjacent to the *pannier* locus (lower bars). Upper right panel, the candidate genomic regions responsible for the major four *h* alleles. Lower bars, the three genomic scaffolds adjacent to the *pannier* locus. Grey, genes predicted from RNA-seq. Green, the *pannier* locus. Contiguity of the scaffolds was predicted based on the truncated genes at the end of scaffolds (*Bgb* and *pnr* (*pannier*)). Arrows indicate the respective candidate genomic regions responsible for each allele. *pannier* is included in all four relevant regions. The images of ladybird beetles are adapted from Fig. 1b. **b** Fold changes of gene expression in the presumptive red and black elytral epidermis around the candidate genomic region responsible for *h*. Grey and red bars on the bottom indicate predicted genes. Red, FDR < 0.01. Only *pannier* was significantly upregulated in this region at 72 h AP. The samples for RNA-seq were collected as depicted in the right panel in the $h^C$ background. **c** Gene structures of *pannier* in *H. axyridis*. Exon number is indicated above each exon. 1A isoform cDNA was cloned by rapid amplification of cDNA ends (RACE). 2A−4B isoforms were predicted from RNA-seq analysis. *pannier* has at least four transcription start sites. Coding sequences are located from exon 2 to exon 5 (yellow). There are two alterative exons at exon 3 (3A and 3B), one encoding one of the two zinc finger domains of Pannier (A isoforms), and the other skipping that zinc finger domain (B isoforms)

*axyridis*. The black-to-red switching phenotype was also observed in *C. septempunctata* adults treated with larval RNAi targeting *pannier* (Fig. 3a, *C. septempunctata*). These data suggest that the elytral colour-patterning function of *pannier* may be conserved at the inter-genus level in ladybird beetles. To investigate the putative regulatory sequences at the *pannier* locus, we performed de novo assembly of the *C. septempunctata* genome using a

linked-read sequencing platform (10× Genomics Chromium system), and obtained a contiguous genomic scaffold including the *pannier* locus (Fig. 5a, *C. sep*) (2.41 Mb; Supplementary Figure 4d; Supplementary Data 2, *C. septempunctata*). Whereas the noncoding sequences of *C. septempunctata pannier* are enriched with several species-specific DNA-binding motifs (Table 1, *C. sep*), we found DNA-binding motifs commonly enriched between

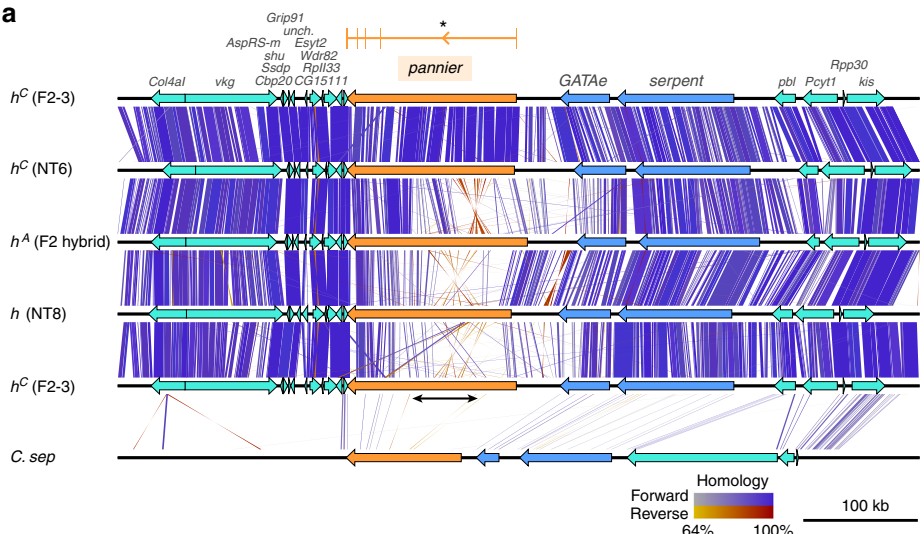

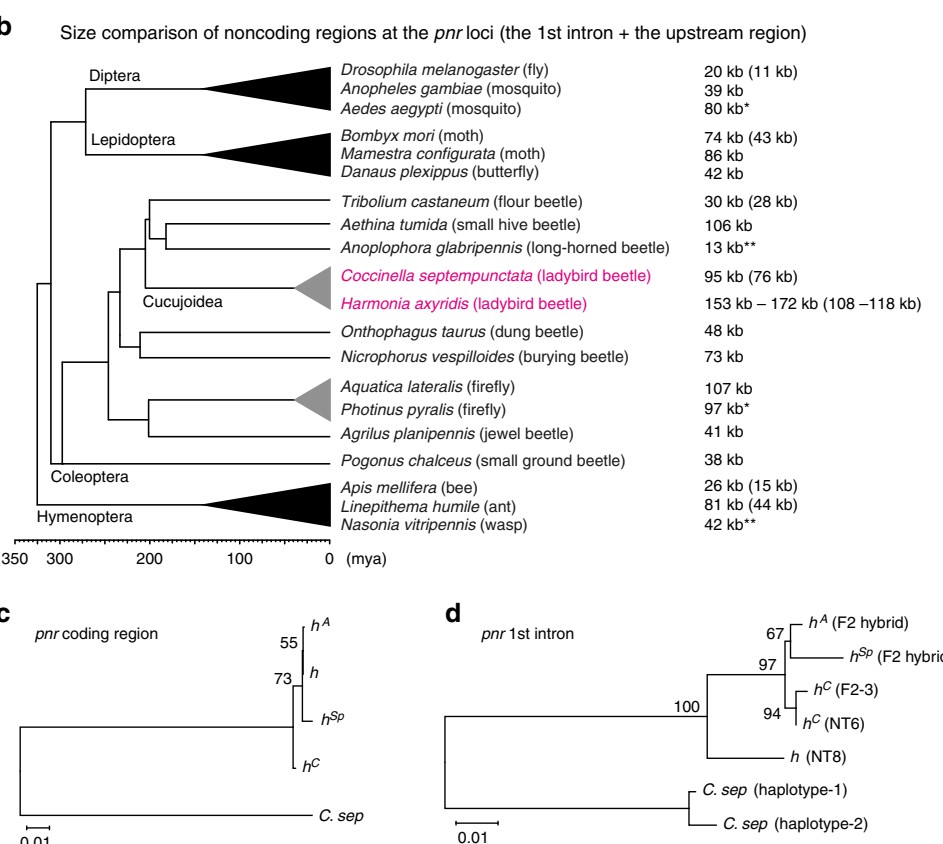

**Fig. 5** Traces of inversions and high sequence diversification within a *pannier* intron in ladybird beetles. **a** Sequence comparison of the genomic region surrounding the *pannier* locus. 700 kb genomic sequences surrounding the *pannier* locus were extracted from the genome assembly of each allele in *H. axyridis* (*h^C^*, *h^A^*, *h*) and *C. septempunctata* (*C. sep*). Strain names are given in parentheses. Arrows indicate genes predicted by the exonerate program (Orange, *pannier*; Blue, GATA transcription factor genes paralogous to *pannier*; Green, other genes). Gene names are listed at the top. Vertical or diagonal bars connecting adjacent genomic structures indicate BLAST[72] hit blocks (bluish, forward hit; reddish, reverse hit) in the comparison between the two adjacent genomic scaffolds. The colour code for colouring the bars is at the bottom. The exon−intron structure and the first intron (*) of *pannier* (1A isoform) is depicted on the top of the panel. The upper half (first intron) of the *pannier* locus is diversified (whitish) between different *h* alleles in *H. axyridis*, and shows traces of inversions (crossed reddish bars). Several intronic sequences are conserved between *H. axyridis* and *C. septempunctata* (bars located in the upper half of *pannier* in *C. sep*). The black arrow indicates the region specifically expanded in *H. axyridis*. **b** Overview of the size of the upper noncoding regions (the first intron + the upper intergenic region) at the *pannier* (*pnr*) locus in holometabolous insects. The topology of the phylogenetic tree of surveyed insects is adapted from ref. [107] (Coleoptera), and TIMETREE[108] (Diptera, Hymenoptera, Lepidoptera). The sizes of the first introns are given in parenthesis if cDNA information was available. *H. axyridis* has the largest noncoding sequence at the *pannier* locus. In some species, synteny of the three paralogous GATA genes was broken up by translocation (*) or insertion (**). **c**, **d** ML phylogenetic trees constructed with nucleotide sequences of *pannier* (*pnr*) coding region (**c**), and those of the conserved region in the first intron (**d**). The trees were drawn to scale with branch lengths measured in the number of substitutions per site. Bootstrap values were calculated from 1000 resampling of the alignment data. Bars, 0.01 substitutions/site

**Table 1 Known DNA-binding motifs enriched in the noncoding regions of the _pannier_ locus**

| Category | Allele/ species | Upstream intergenic region | Upstream region of the first intron | Downstream region of the first intron |
|---|---|---|---|---|
| Allele-/species-specifically enriched motifs | $h^C$ | EcR, Foxo | En, Exd, H, HLH106, Mad, Myc, Mnt, Ss, Pad, Pan, Poxn, Sd, Tgo, Tai, Vvl | Ab, B-H1, Crc, Crol, Dr, Rn, Sd |
| | $h^A$ | Exd, Pan, Vvl | Ab, Brk, Pnr, Spps, Usp, | Usp |
| | $h$ | Kni, Rn, Sqz | Ab, Ato, Ets21C, Rn, Sqz | Al, B-H1, Lms, Nub, Sqz, Tup |
| | C. sep | Ato, EcR, H, Ss, Tgo | B-H2, Eg, Pnr, Tai, Tgo | Brk, E(spl)mβ, H, Ken, Tai, Tgo |
| Commonly enriched motifs | H. axy | Ab, Brk, Mad, Taxi, Tgo | Al, Brk, E(spl)m E(spl)mβ, Exd, Hth, Lms, Mad, Tai, Tgo, Tx | Dr, En, Inv, Med, Sens, Slou, Unpg |
| | H. axy & C. sep | Mad | Exd, Hth, Lms, Mad, Tai, Tgo | — |
| Region size (bp) | $h^C$ | 38,192 (F2-3)/51,566 (NT6) | 71,107 (F2-3)/76,264 (NT6) | 47,450 (F2-3)/44,236 (NT6) |
| | $h^A$ | 42,826 | 75,417 | 52,210 |
| | $h$ | 40,778 | 67,340 | 67,340 |
| | C. sep | 20,370 | 56,564 | 18,432 |

Enriched DNA-binding motifs of _Drosophila_ transcription factors involved in wing formation are listed ($p < 0.05$). The Scalloped binding motif (Sd) discussed in the text is underlined.

_Harmonia_ and _Coccinella_, which are associated with wing vein formation and wing/body wall patterning (Exd, Hth and Mad)[29,31,32] (Table 1, commonly enriched motifs). Therefore, co-option of such wing developmental modules in the regulatory region may have facilitated acquisition of a novel expression domain of _pannier_ in pupal elytral blades in ladybird beetles.

In order to explore the history of the emergence of elytral colour patterns in _H. axyridis_, we also performed a molecular phylogenetic analysis focusing on the highly conserved _pannier_ intronic sequences shared among _H. axyridis_ alleles and _C. septempunctata_ (three blocks, totalling 1.1 kb in length, Supplementary Data 5). The maximum likelihood (ML) phylogenetic tree inferred from nucleotide sequences of the _pannier_ coding region did not resolve the phylogenetic relationship among the alleles in _H. axyridis_ to a satisfactory level (Fig. 5c, bootstrap values <75). However, the ML tree inferred from the conserved intronic sequence suggested that in _H. axyridis_ the contrasting colour patterns of the _h_ allele (black spots in red background) and the other three alleles (red spots in black background) diverged first. The latter three alleles diverged more recently (Fig. 5d, bootstrap values >90).

## Discussion

The _pannier_ locus identified in this study appears to be the key genetic locus responsible for the origin of large-scale intraspecific variation genetically linked to the _h_ locus in ladybird beetles[1,2]. Also, it is worth noting that a concurrent study by Prud'homme, Estoup and their colleagues independently identified the same locus in _H. axyridis_ by whole-genome sequencing, population genomics, gene expression and functional genetics approaches[37]. Based on the results presented in this study, we propose an evolutionary model that might underlie the high level of diversification of the intraspecific elytral colour patterns of _H. axyridis_. In addition, we also discuss the underlying evolutionary developmental backgrounds specific to ladybird beetles.

The common ancestor of _Harmonia_ and _Coccinella_ (Coccinellinae) diverged more than 33.9 million years ago, according to molecular phylogenetic analyses and fossil records[38,39]. Therefore, the elytral colour-patterning function of _pannier_ shared between _H. axyridis_ and _C. septempunctata_ was most likely acquired before this divergence event. The 1.1 kb sequence blocks in the first intron of _pannier_ conserved between _H. axyridis_ and _C. septempunctata_ are a likely candidate for a regulatory element associated with the ladybird beetle-specific elytral expression of _pannier_ in the pupal elytra. The effects of enhancer activities of

these sequence blocks have not yet been experimentally addressed. However, the acquisition of such regulatory sequences during evolution would have coincided with the acquisition of the elytral colour-patterning function of _pannier_ (Fig. 6, blue diamond). These conserved sequence blocks are located in the expanded intronic region specific to _H. axyridis_ (Fig. 5a, black arrow). Therefore, the expansion of the first intron in the ancestral lineage of _H. axyridis_ (Fig. 6, intronic expansion) might be one of the events that facilitated diversification of the intraspecific elytral colour patterns.

In the genus _Harmonia_, colour patterns similar to those encoded by the _h_ allele and those of _C. septempunctata_ (black spots in red background) are commonly observed. Also, the position of the spots is similar across species (e.g. _H. quadripunctata_, _H. octomaculata_, and _H. dimidiata_). Therefore, we speculate that the intronic sequence of _pannier_ in the _h_ allele of _H. axyridis_ might retain a repertoire of regulatory sequences acquired in a common ancestor of the genus _Harmonia_ (Fig. 6, green arrowhead). However, in the ancestral lineage of _H. axyridis_, the regulatory region of _pannier_ appears to have been modified to generate novel colour patterns of the recently diverged alleles ($h^C$, $h^{Sp}$ and $h^A$; red spots in black background; Fig. 6 magenta, red and purple arrowheads). The 70 kb-scale noncoding sequences located at the upstream region of the first intron of _pannier_ that is specifically expanded in _H. axyridis_ (Fig. 6, Intronic expansion, yellow box) might have facilitated accommodation of the allele-specific regulatory motifs responsible for the diversified colour pattern of elytra. In addition, traces of inversions in this region consistently found in allele comparisons suggest that repeated inversions in this region (Fig. 6, white arrowheads) created opportunities to diverge the noncoding sequence of _pannier_ to successively generate novel diverse alleles within a species by suppressing recombination within this region. Such inversion events would have occurred in the common ancestor of _H. axyridis_ and its reproductively isolated sister species, _H. yedoensis_ because the major elytral colour patterns are shared between the two species[40]. Large-scale chromosomal inversion is believed to be one of the major driving forces generating and maintaining intraspecific morphological variation within a species[41–44]. Our study exemplifies that not only a single inversion event but also repeated inversion events at an expanded intron can lead to the acquisition of novel morphological traits within a species.

From the viewpoint of evolutionary developmental biology, it is noteworthy that in _H. axyridis_, of all of the developmental genes known to regulate colour pattern and pigmentation, a

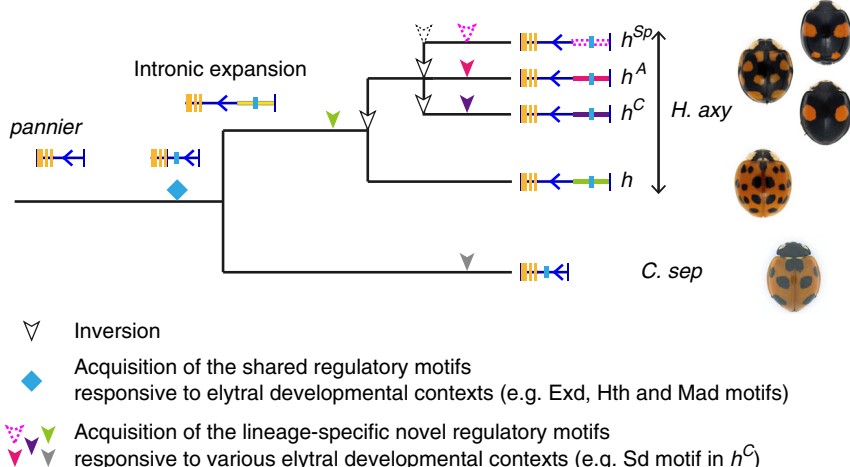

**Fig. 6** An evolutionary model for the colour pattern diversification in *H. axyridis*. See details in the Discussion. The images of ladybird beetles are adapted from Figs. 1b and 3c

single gene, *pannier*, is responsible for the major classes of intraspecific entire wing colour pattern diversification. This evolutionary pattern contrasts with that of the intensely studied warning signals of *Heliconius* butterflies. In the case of *Heliconius erato* and *Helconius melpomene*, five major loci and several minor loci located on different chromosomes regulate multiple intraspecific wing colour patterns prevailing in the population[45]. This difference in evolutionary mechanisms may stem from a paucity of available options of evolvable genes in the gene regulatory network of elytral colour patterning. Ladybird beetles diverged from ancestral species of Cucujoidea[38] (Fig. 5b, Cucujoidea), leaf-litter or rotten-tree dwelling insects. Thus, the ancestor of ladybird beetles would have had far less colourful and more simply patterned forewings (elytra) than the ancestors of butterflies, moths. Therefore, these ancestors presumably would have possessed far fewer colour pattern regulatory genes. In *H. axyridis*, this developmental constraint may have led to the selection of *pannier* as the major evolvable gene to a signal-integrating "input−output" regulatory gene[46,47]. This might have generated >200 colour patterns genetically tightly linked to the *h* locus by utilising the expanded regulatory DNA sequence. Future research aiming to identify specific regulatory inputs to *pannier* will help clarify the regulatory mechanisms underlying the generation of highly diverse intraspecific polymorphism at the interspecific level. Another important issue to clarify whether *pannier* is indeed the hotspot of morphological evolution in ladybird beetles is whether *pannier* is responsible for the remaining >20 minor colour patterns in *H. axyridis*.

## Methods

**Insects**. Laboratory stocks of *H. axyridis* and *C. septempunctata* were derived from field collections in Japan. They were reared at 25 °C and usually fed on artificial diet[48], or fed on the pea aphid *Acyrthosiphon pisum* (kindly provided by Dr. T. Miura) for egg collection. Larvae and pupae analysed in this study were not sexed.

**Phenoloxidase (PO) activity staining**. Pupa elytral discs were dissected in a potassium phosphate buffer (K-$PO_4$ buffer; 100 mM $KH_2PO_4$/$K_2PO_4$, 150 mM NaCl, pH 6.3) on ice. PO staining was performed using 0.4 mg/ml dopamine as a substrate in 40% K-$PO_4$ buffer/60% isopropyl alcohol for 2 h at room temperature as previously described[12]. After washing several times in the potassium phosphate buffer containing 0.3% Triton-X100 and mounted in this solution. Images were captured with a stereoscopic microscope (MZ FLIII, Leica) equipped with a digital camera (DP70, Olympus).

**Histological analysis**. To visualise tissue morphology and PO active tissues, pharate adult elytra dissected in ice-cold PBS (137 mM NaCl, 2.68 mM KCl, 10.14 mM $Na_2HPO_4$, pH 7.2) at 96 h AP or those after PO activity staining were fixed

with 4% paraformaldehyde (PFA) in PBS for 15 min on ice and for 75 min at room temperature. After fixation, the elytra were washed several times in 100% methanol and stored in 100% methanol at −20 °C until use. After dehydration, the elytra were embedded in 4% carboxymethyl cellulose (FINETEC) and were frozen in hexane cooled with dry ice. The freeze-embedded elytra were stored at −80 °C until use. The 6 μm frozen sections were prepared using an adhesive film (Cryofilm Type 1; FINETEC)[49]. Sections of the PO activity-stained elytra were dried at least 1 h at room temperature, mounted in PBS, and photographed under an inverted microscope (IX70, Olympus). For nuclear and F-actin staining, sections were treated with 2.5 μg/ml propidium iodide, 1 mg/ml RNase A and 5 U/ml AlexaFluor 488 phalloidin (Molecular Probes) for 1 h at 37 °C under a dark condition. After washing three times in PBS, the sections were mounted in an antifade reagent (FluoroGuard™; Bio-Rad), and images were captured with a confocal laser-scanning microscope (LSM 510; Carl Zeiss).

For localisation of carotenoid, elytra at 96 h AP were embedded and sectioned as described above. All procedures were rapidly performed to prevent diffusion of carotenoids. The sections were dried for 1 min, mounted in PBS and immediately photographed under an inverted microscope (IX70, Olympus).

**cDNA cloning**. Larval and pupal elytral discs and pharate adult elytra of *H. axyridis* ($h^C$) and *C. septempunctata* were dissected in PBS on ice. Soon after dissection, the tissues were frozen in liquid nitrogen and stored at −80 °C until use. Total RNA was extracted from each sample using TRIzol Reagent (Invitrogen) or RNeasy Micro Kit (Qiagen) according to the manufacturer's instructions, and treated with 2 U DNase I (Ambion) for 30 min at 37 °C. The first-strand cDNA was synthesised with SMARTer PCR cDNA Amplification Kit (Clontech) using 1 μg of total RNA according to the manufacturer's instructions. *H. axyridis* and *C. septempunctata* cDNA fragments were amplified by reverse transcription-polymerase chain reaction (RT-PCR) and rapid amplification of cDNA ends (RACE) with the primers listed in Supplementary Tables 2, 3. The PCR product was cloned into the *EcoR* V site of the pBluescript KS + vector (Stratagene) or pCR4-TOPO vector (TOPO TA Cloning Kit; Invitrogen). The nucleotide sequences of the PCR products were determined using a DNA sequencer 3130 genetic analyser (Applied Biosystems). The SNPs in open reading frame (ORF) of *pannier* were determined through direct sequencing of the PCR products treated with ExoSAP-IT (Affymetrix). Sequencing was performed by DNA sequencing service (FASMAC) using the primers listed in Supplementary Table 4. Sequence analysis was carried out using DNASIS (Hitachi Software Engineering) or ApE[50] (version 2.0.45) software. Nucleotide sequences and deduced amino acid sequences were aligned with ClustalW in MEGA[51] software (version 7.0.18). The alignment figures were generated using Boxshade[52] (version 3.21).

**Gene expression analysis by RT-PCR**. For the gene expression analysis in each developmental stage, elytral tissues of three individuals of *H. axyridis* ($h^C$) and *C. septempunctata* were dissected as described above. Six elytral tissues from each sampling stage were pooled in one test tube. Total RNA extractions and the subsequent first-strand cDNA syntheses (using 425 and 267 ng of total RNA for *H. axyridis* and *C. septempunctata* samples, respectively) were performed as described above. Three microlitres of 100 and 62.8 times diluted *H. axyridis* and *C. septempunctata* first-strand cDNA was used as a template for each PCR, respectively. The PCR cycle number was 35 for all genes. A set of primers #1 and #2 for each gene was used for this analysis (Supplementary Table 5).

For the gene expression analysis in the future red and black regions, the red and black regions of pharate adult elytra at 84 h AP were collected by boring with

injection needles. Internal diameters of the needles were 0.7 and 0.6 mm for *H. axyridis* and *C. septempunctata*, respectively. In the case of *C. septempunctata*, elytra stained with PO activity were used for boring because carotenoid localisation was not observable unlike in *H. axyridis*. cDNA synthesis was performed as described above, using as much total RNA as we could extract. Twenty microlitres of ten times diluted first-strand cDNA was used as a template for PCR. The PCR cycle numbers were 45 cycles for *Ha-pnr*, 38 cycles for *Ha-rp49*, 47 cycles for *Cs-pnr* and 40 cycles for *Cs-rp49*. A set of primers #3 and #4 for each gene was used for this analysis other than *rp49*. *Ha-rp49* and *Cs-rp49* were used as internal controls. Reactions without reverse transcriptase were performed with cDNA synthesis as negative control samples for the RT-PCR experiments. No band was detected in these reactions for all genes. The primers used for this analysis are described in Supplementary Table 5.

**Larval RNAi.** DsRNA synthesis and microinjection into larvae were performed as described previously[19]. In brief, the cloned cDNA fragments in DNA vectors were amplified by PCR using the primers flanked with the T7 promoter sequence (Supplementary Table 6), and used as templates for dsRNA synthesis. Amplified PCR products were separated by electrophoresis on 1% agarose gels and purified using MagExtractor PCR & Gel Clean up kit (Toyobo). DsRNAs were synthesised using the MEGA script T7 kit (Ambion). Approximately 1.4−2.7 μg and 1.4−2.0 μg of the dsRNAs of *Ha-pnr* and *Cs-pnr* were injected into 2-day-old forth (final) instar larvae, respectively. Approximately 2.0−2.7 μg and 1.4−2.7 μg of the *EGFP* dsRNA were injected into *H. axyridis* and *C. septempunctata* larvae as negative controls, respectively. For other genes in the initial small screening, approximately 1 μg of dsRNA was injected into each early last instar larva. Different amount of dsRNA for each gene in this range gave no difference in phenotypic effects. In order to give enough time for the completion of pigmentation, images of adults were captured more than 2 days after eclosion using a digital microscope (VHX-900, Keyence).

**In situ hybridisation.** Essentially the same protocol for whole mount pupal antennal primordia of the silk moth[53] was used. For sclerotized pupal elytra of ladybird beetles, several procedures were modified as follows: to increase RNA probe penetrance in elytral epidermis covered with sclerotised cuticle at 76−84 h AP, the peripheral edge of an elytron was cut off, and then, ventral and dorsal elytral epidermis layers appressed together were carefully separated with fine forceps after fixation; to reduce nonspecific probe hybridisation, fixed, separated and detergent-permeabilised elytra epidermal samples were stored in 100% methanol for more than 12 h at −30 °C, and prehybridisation treatment was extended to two overnight incubation; the concentration of cRNA probes were reduced to 0.4 ng/μl; the ventral epidermis samples were not used for analysis because of high non-specific background signals. Sample washing was performed for 10 min three times unless otherwise noted.

*pannier* antisense probes were designed at 5′ and 3′ regions of ORF excluding the two conserved GATA zinc finger coding regions in the middle to prevent cross-hybridisation with other GATA family genes. The PCR primers used to amplify the template DNA for in vitro RNA probe synthesis were listed in Supplementary Table 6. Briefly, *pannier* ORF fragment was amplified by RT-PCR using cDNA from 72 h AP ($h^C$), and cloned into pCR4-TOPO vector (Invitrogen). Sense and antisense probe templates were amplified from the cloned cDNA. Sense and antisense DIG-labelled riboprobes were transcribed using the flanking T7, T3 or SP6 promoter sequences, and DIG RNA labelling kit (Roche). Mixture of 5′ and 3′ probes was used for hybridisation. The concentrations of RNA probes were quantified by agarose gel electrophoresis.

First, pupal elytra were dissected in PBS, and fixed with 4% PFA in PBS for 2 h, and washed with PBS including 0.1% Tween20 (PTw). Fixed elytra were dorsoventrally separated using fine forceps, and permeabilized with 0.5% Triton X-100 for 45 min. After washing with PTw, samples were treated with 20 μg/ml proteinase K in PTw at 37 °C for 30 min. Proteinase K was immediately washed out by quick washes with 2 mg/ml glycine in PBS and following washes with PTw. After postfixation with 4% PFA and 0.1% glutaraldehyde for 30 min, samples were equilibrated to hybridisation solution with five steps of hybridisation solution wash series (50, 75, 87.5, 100, 100%), and prehybridized at 57 °C over two nights (for ca. 40 h), and hybridised with 0.4 ng/μl cRNA probes overnight. After reverse hybridisation solution wash series (50, 25, 12.5, 0% (PTw) 0% (PTw)) and a wash with RNase A reaction buffer (10 mM Tris-HCl, 500 mM NaCl, pH 8.0), single stranded probes not hybridised to mRNA were degraded with 20 μg/ml RNase A at 37 °C for 30 min. After equilibrated to hybridisation solution again, nonspecifically bound degraded probes were washed out with 4 times of 20 min washes with hot hybridisation solution at 57 °C and following hot reverse hybridisation solution wash series (50, 25, 12.5%; 20 min for each). After washing with PBTw and PBSS (PBS with 0.01% Saponin), and blocking for 30 min with PBSS-BSA (PBSS with 0.2% bovine serum albumin), samples were incubated with alkaline phosphatase-conjugated anti-DIG Fab fragment (1:2000, Roche) in PBSS-BSA. After washing with PBSS and NTMT buffer (0.1 M NaCl, 0.1 M Tris-HCl (pH 9.5), 50 mM MgCl₂, 0.1% Tween20), Colour development was conducted using 170 μg/ml BCIP (Roche) and 340 μg/ml NBT (Roche) diluted in NTMT buffer. After washing with NTMT buffer and PBSS, the solution was exchanged to 100% ethanol in two steps (50, 100%), and the samples were de-coloured for 1 h. After returning to PBSS in two steps, the samples were mounted in 80% glycerol, and observed under a

stereomicroscope (Stemi 508, Zeiss). Images of the samples were collected using a digital camera system (NY-D5500 super system, Microscope Network). Brightness and contrast of each image was adjusted using Photoshop CS6 (Adobe). The same image processing was applied to all images.

**De novo genome assembly of *H. axyridis*.** A single female adult from F2-3 strain sibcrossed for three generations ($h^C$) was used for the first version of de novo genome assembly of *H. axyridis*. Genomic DNA was extracted using DNeasy Blood and Tissue Kit (Qiagen). Paired-end (300 and 500 bp) and mate pair (3, 5, 8, 10, 12 and 15 kb) libraries were constructed using TruSeq DNA PCR-Free LT Sample Prep Kit and Nextera Mate Pair Sample Prep Kit (Illumina) following the manu-facturer's protocols. Sequencing libraries were run on Illumina HiSeq2500 sequencers. In total, we generated 133.6 Gb of raw sequence data for de novo genome assembly. Genome assembly was performed using the Platanus v1.2.1.1 assembler[54] after removal of adapter sequences and error correction (SOAPec v2.01)[55].

**Reassembly of the genomic scaffold at *H. axyridis pannier*.** Adaptor sequences and low-quality regions in paired-end and mate-pair reads were trimmed using Platanus_trim[56] (version 1.0.7) with default parameters. Trimmed reads were assembled by Platanus2[57] (version 2.0.0), which was derived from Platanus[54] to assemble haplotype sequences (i.e. haplotype phasing) instead of consensus sequences. Procedures of Platanus2 are briefly described as follows: (1) De Bruijn graphs and scaffold graphs are constructed without removal of bubble structures caused from heterozygosity. Paths that do not contain junctions correspond to assembly results (scaffolds). Scaffold pairs in bubbles represent heterozygous haplotypes. (2) Paired-ends or mate-pairs are mapped to the graphs to detect links between bubbles, and linked bubbles are fused to extend haplotype sequences. (3) Each haplotype (contig or scaffold) is independently extended by modules of de novo assembly derived from Platanus. (4) Homologous pairs of haplotype scaffolds are detected using bubble information in the initial de Bruijn graph. (5) Steps 1−4 are iterated using various libraries (paired-ends or mate-pairs). (6) Homologous pairs of scaffolds are formatted into bubble structures as output. For each pair, longer and shorter scaffold were called "primary-bubble" and "secondary-bubble", respectively. Primary-bubbles, secondary-bubbles and nonbubble scaffolds are collectively called "phased-scaffolds".

In addition, Platanus2 can connect primary-bubbles and nonbubble scaffolds to construct long "consensus scaffolds", which consists of mosaic structure of haplotypes (i.e. paternal and maternal haplotypes are mixed). Employing the strategy of Platanus2, certain highly heterozygous regions were expected to be assembled contiguously compared to Platanus.

Using the markers of the responsible region for elytral colour patterns (the *h* locus), we found that two long bubbles and one short nonbubble scaffold corresponded to the locus. Consequently, one consensus scaffold covering the breakpoint markers at the *h* locus was constructed from these phased scaffolds. We used that consensus scaffold (3.13 Mb) for the downstream in silico sequence analyses. We assessed the completeness of the genome assembly using BUSCO[58] (version 3.0.2, Insecta dataset (1658 orthologues)).

**Genome sequencing by long reads and linked-reads.** High molecular weight (HMW) genomic DNA was extracted using QIAGEN Genomic-tip 100/G (QIAGEN) according to the manufacturer's instructions. The concentrations and qualities of the extracted HMW genomic DNA were evaluated using Qubit dsDNA, and RNA HS kits (Thermo Fisher).

For library preparation for 10× Genomics Chromium system, one pupa ($h^C$ (NT6 strain) and *h* (NT8 strain)) or one adult ($h^A$ (F2 adult progenies in genetic cross $h^A \times h^{Sp}$) and *C. septempunctata* (MD-4 strain)) was used. Size selection by BluePippin (range: 50 kb−80 kb, Sage Science) was performed only for $h^A$ genomic DNA used in 10× linked-read library preparation.

Preparation of gel bead-in-emulsions (GEMs) for each 10× Genomics Chromium library was performed using 0.5–0.6 ng of HMW genomic DNA according to the manufacturer's instructions. The prepared GEMs were quality-checked using Qubit dsDNA HS kit (Thermo Fisher) and Bioanalyzer (Agilent), and processed with Chromium Controller (10× Genomics). The constructed DNA libraries were quality-checked again in the same way. Sequencing of the libraries was performed in the Hiseq X ten (Illumina) platform (1 library/lane) at Macrogen. In total, we generated 66.9, 64.6, 64.9 and 60.3 Gb of raw reads for linked-read sequencing ($h^C$, $h^A$, *h*, and *C. septempunctata*, respectively).

For library preparation for PacBio system, 10−11 pupae ($h^C$ (NT6 strain) and *h* (NT8 strain)) or adults ($h^A$ (F2 adult progenies in genetic cross $h^A \times h^{Sp}$)) were used. The libraries were prepared according to the 20-kb Template Preparation Using BluePippin™ Size-Selection System (Sage Science). Sequencing of the libraries was performed in the PacBio RS II (Pacific Biosciences) platform. In total, 4.31, 4.92 and 4.44 Gb of insert sequences (approximately 10× coverage of the genome, assuming a genome size of 423 Mb) were obtained from 4 to 5 SMRT cells for $h^A$, $h^C$ and *h*, respectively.

**De novo assembly of 10× linked-reads.** For 10× linked-reads libraries of four samples (three *H. axyridis* and one *C. septempunctata*), Supernova (version 2.0.0)[59]

was executed with default parameters except for the maximum number of used reads (the --maxreads option) to obtain the optimum coverage depth for Supernova (56×). For each sample, the value for --maxreads was determined as follows: (1) Barcode sequences in raw linked-reads were excluded using "longranger basic" command of Long Ranger[60] (version 2.1.2), resulting in "barcoded.fastq" file. (2) Adaptor sequences and low-quality regions in "barcoded.fastq" were trimmed using Platanus_trim (version 1.0.7) with default parameters. (3) 32-mers in the trimmed reads were counted by Jellyfish[61] (version 2.2.3) using the following two commands and options:

$ jellyfish count -m 32 -s 20M -C -o out.jf barcoded_1.trimmed barcoded_2.trimmed

$ jellyfish histo -h 1000000000 -o out.histo out.jf

In summary, all 32-mers in both strands (-C) were counted and distribution of the number of occurrences without upper limit of occurrences (-h 1000000000). (4) The haploid genome size was estimated using the custom Perl script. For the distribution of the number of 32-mer occurrences ("out.histo"), the number of occurrences corresponding to a homozygous peak was detected, and the total number of 32-mers was divided by the homozygous-peak-occurrences. Here, 32-mers whose occurrences were small (<the number of occurrences corresponding to the bottom between zero and heterozygous peak) were excluded for the calculation to avoid the effect from sequencing errors. (5) The values for --maxreads were calculated as follow:

estimated-haploid-genome-size / mean-read-length-of-barcoded.fastq × 56

As a result, we obtained the scaffolds including the genes surrounding *H. axyridis-pannier* ($h^C$ (NT6), 2.74 Mb; $h^A$ (F2 hybrid), 1.42 + 1.61 Mb; $h^C$ (NT8) 2.79 Mb), and homologous regions in *C. septempunctata* (haplotype 1, 10.16 + 2.41 Mb; haplotype 2, 10.13 + 2.44 Mb). We used those sequences for the downstream in silico analyses. We assessed the completeness of the genome assembly using BUSCO[58] (version 3.0.2, Insecta dataset (1658 orthologues)).

**Gap filling of the genomic scaffolds at the *pannier* locus.** Concerning the genome assemblies of *H. axyridis*, we used minimap2[62] (ver. 2.9) and PBjelly[63] (ver. PBSuite_15.8.24) software to fill gaps around the *pannier* locus. In each genome of three strains of *H. axyridis*, we first mapped PacBio reads to the genome assemblies generated from the 10× linked-reads using minimap2. Then, we chose PacBio reads mapped to the scaffold containing *pannier* gene. These PacBio reads were subjected to gap-filling of the scaffold with PBjelly. We obtained gap-free nucleotide sequences spanning the entire *pannier* locus and the upstream intergenic regions.

Concerning the genome assembly of *C. septempunctata*, there was a single gap estimated to be 15 kb long by Supernova program in the first intron of *pannier* locus. We handled this gap region as repeated *N*, and included it in the downstream in silico analyses.

**Validation of the *pannier* scaffold re-assembled by Platanus2.** For the *H. axyridis* F2-3 sample, trimmed reads of the 15 kbp-mate-pair library were mapped to the consensus scaffold set of Platanus2 using BWA-MEM[64] (version 0.7.12-r1039) with default parameters. Next, a consensus scaffold corresponding to the *pannier* locus was segmented into 2 kbp-windows, and links between windows (≥3 mate-pairs) were visualised by Circos[65] (version 0.69-6).

**Preliminary resequencing of *H.axyridis* genome for RAD-seq.** Genomic DNA was extracted from each of $h^C$ (F6 strain), $h^A$ (NT3 strain) and $h^{Sp}$ (CB-5 strain), and used to create Illumina libraries using TruSeq Nano DNA Sample Preparation Kit (Illumina) with insert size of approximately 400 bp. These libraries were sequenced on the Illumina HiSeq 1500 using a 2 × 106-nt paired-end sequencing protocol, yielding 84.7 M paired-end reads. SNP site identification was conducted basically according to the GATK Best Practice[66] (ver. August 7, 2015). After trimming adaptor sequences with Cutadapt software (ver. 1.9.1)[67], the sequence data were mapped to the de novo genome assembly data using bwa software (ver.0.7.15, BWA-MEM algorithm)[64]. Sequences and alignments with low quality were filtered using Picard[68] tools (ver. 2.7.1) and GATK[66] software (ver. 3.6 and 3.7), and 734,443 SNP markers in the strains were identified. The most distantly related strains ($h^C$ (F6 strain) and $h^A$ (NT3 strain)) were selected for the RAD-seq analysis by performing phylogenetic analysis using SNPhylo[69] (Version: 20140701).

**Comparison of the genomic scaffolds at *pannier* in ladybirds.** For each pair of the entire scaffolds and the extracted *pannier* region, we constructed dot plots by performing pairwise-alignment using "nucmer" program in the MUMmer package[70] (version 3.1). The options of nucmer were as follows: (1) the entire scaffolds, *H. axyridis* vs. *H. axyridis*, Default parameters; (2) the entire scaffolds, *H. axyridis* vs. *C. septempunctata*, "-l 8 -c 20"; (3) the *pannier* region, *H. axyridis* vs. *H. axyridis*, "-l 12". Alignment results (delta files) were input into "mummerplot" program to generate dot plots. Note that resultant gnuplot scripts resulting from mummerplot were edited for visualisation.

We also visualised the homology and structural differences between the 700 kb-genomic region including *pannier* using Easyfig[71] (ver. 2s2.2). Short BLAST[72] hit fragments less than 500 bp, and putative short repeat sequences less than 1250 bp, which showed more than two BLAST hit blocks within the 700 kb region, were

filtered using a custom Perl script. Exon−intron structures of putative genes in the 700 kb regions were obtained using Exonerate[73] (ver. 2.2.0) with the options "-m est2genome --showvulgar yes --ryo">%qi length=%ql alnlen=%qal/n>%ti length=%tl alnlen=%tal/n" --showtargetgff yes --showalignment no --score 2000′. cDNA sequences cloned by RT-PCR or predicted by RNA-seq were used as queries. If a single cDNA unit was split into multiple fragments, we merged the fragments by performing exonerate search again using the cDNA sequences whose subsequences were substituted by the genomic hit fragments in the first exonerate search as a query. Exonerate output files were converted to the GFF3 format using our bug-fixed version of the process_exonerate_gff3.pl[74] Perl script with the option "–t EST". The GFF3 file and a FASTA format file of each scaffold were converted to a GENBANK format file using EMBOSS Seqret[75] program (ver. 6.6.0.0) with the options "-fformat gff -osformat genbank". The GENBANK format files corresponding to the 700 kb genomic sequences surrounding *pannier*, which were used as input files of Easyfig, were extracted using the Genbank_slicer.py[76] Python script (ver. 1.1.0).

**Flexible ddRAD-seq.** We newly constructed a flexible ddRAD-seq library preparation protocol to facilitate high-throughput ddRAD-seq analyses at low cost. We designed all enzymatic reactions to be completed sequentially without DNA purification in each step to make the procedures simple. In addition, we designed 96 sets of indexed and forked sequencing adaptors compatible with Illumina platform sequencers (Supplementary Data 6).

Briefly, 100 ng of genomic DNA was first double-digested with 15 U of *Eco*RI-HF and 15 U of *Hind*III-HF in 20 µl of NEB CutSmart Buffer (New England Biolabs) at 37 °C for 2 h. Fifteen microlitres of the digested DNA, 4 pmol of adaptor DNA, 10 µmol of ATP, 400 U of T4 DNA ligase were mixed in 20 µl, incubated at 22 °C for 2 h, and denatured at 65 °C for 10 min. Ligated library DNA fragments were purified with Agencourt AMPureXP (Beckman Coulter) according to the manufacturer's instructions. Library DNA fragments ranging from 300 to 500 bp were size-selected with Pippin Prep (Sage Science). Concentration of each library DNA was quantified using KAPA Library Quantification Kits (Roche) according to the manufacturer's instructions. Sequence data were obtained by applying 96 DNA libraries to a single lane of Hiseq 1500 (Illumina).

**Linkage map construction and a genome-wide association study.** A single $h^C$ male (F6 strain) and a single virgin $h^A$ female (NT3 strain) were crossed, and the obtained F1 progenies were backcrossed with the F0 male ($h^C$, F6 strain). Finally, 183 adult F2 progenies ($h^C = 80$, $h^C/h^A = 103$) and 2 F0 adults were collected for RAD-sequence analysis, and stored at −30 °C until use. Genomic DNAs were extracted individually using an automatic nucleic acid extractor (PI-50α, KUR-ABO). Briefly, each frozen ladybird beetle and a zirconia bead were transferred to 2 ml plastic tube (Eppendorf) on ice. Immediately, 250 µl of cold lysis buffer including Proteinase K and RNase A, but not SDS was added to the sample, and the tubes were vigorously shaken with a tissue grinder (Tissue Lyser LT, Qiagen) at 3000 rpm for 1 min. Then, 250 µl of lysis buffer including SDS was added to each crushed sample, and processed with the automatic program for DNA extraction from mouse tail, according to the manufacturer's instructions. The extracted genomic DNA was diluted in 30 µl of TE buffer. The DNA concentration of each sample was quantified using Qubit dsDNA BR Kit according to the manufacturer's instructions (Thermo Fisher Scientific). We performed flexible ddRAD-seq using these genomic DNA samples. 0.6–6.0 million (mean = 2.0 million) of 106 bp paired-end reads per sample were generated using two lanes of Hiseq 1500 (Illumina) following the methods in the User Guide.

Mapping and polymorphic site calling were conducted as described in the resequencing analysis above except that the procedure for filtering duplicated reads using Picard was eliminated because we did not amplify DNA library by PCR. Count data at each SNP sites were extracted from the obtained vcf file using vcf_to_rqtl.py script in rtd software[77] with the options "5.0 80". To avoid program errors, we modified the script to skip the read depth data (DP) including characters in the GATK vcf file.

We constructed a linkage map using R/qtl[78] (version 1.42.8) and R/ASMap[79] (version 1.0.2) R[80] packages according to the QTL mapping workflow for BC1 population of *Jaltomata*[81]. Using the obtained csv file as an input, we eliminated the polymorph sites that behaved as located on the X chromosome. In addition, individuals with low mapping quality, and marker sites with low-quality or highly distorted segregation patterns were eliminated as well. Finally, 4419 markers sites and 177 F2 individuals were used. The linkage map was initially constructed with mstmap program (R/ASMap) with the options "dist.fun = 'kosambi', p.value = 1e-25", and highly linked linkage groups were merged manually. The markers consistently incongruent with neighbouring markers were eliminated using correctGenotypes.py[81] Python script with the options "-i csvr -q 0.1 -t 4.0".

A genome-wide association study (GWAS) was conducted using calc.genoprob program (R/qtl) with the options "step = 1, error.prob = 0.001" and scanone program (R/qtl) with the option "model = 'binary'". The result data were visualised with the "plot" program in R/ASMap.

**Genetic association studies focusing on the *pannier* locus.** In addition to the genetic cross in the GWAS ($h^C × h^A$), two independent crosses (($h × h^C$) and ($h^A × h^{Sp}$)) were performed.

In the former cross, a single $h$ male (D-5 strain) and a single virgin $h^C$ female (F2-3-B strain) were crossed, and the obtained F1 progenies were sibcrossed. Finally, 80 F2 adult progenies ($h^C$ = 30, $h^C/h$ = 34, and $h$ = 16) were collected for genotyping, and stored at −30 °C until use. In the latter cross, a single $h^{Sp}$ male (CB-5 strain) and a single virgin $h^A$ female (NT3 strain) were crossed, and the obtained F1 progenies were sibcrossed. Finally, 273 F2 adult progenies ($h^{Sp}$ = 103, $h^{Sp}/h^A$ = 80, and $h^A$ = 90) were collected for genotyping, and stored at −30 °C until use.

Genomic DNA was extracted individually using the automatic nucleic acid extractor (PI-50α, KURABO) as described in the previous section, and diluted to approximately 100 ng/μl. We searched for genotyping markers by amplifying and sequencing the intronic region of the genes surrounding *pannier* with PCR. The individual PCR was performed using approximately 100 ng of genomic DNA and Q5 DNA polymerase (New England Biolabs) with 45 cycles. The primers used, the markers identified and the typing results are summarised in Supplementary Data 1.

**RNA-seq analysis**. The total RNA extraction procedure for RNA-seq is essentially the same as that for the gene expression analysis in the presumptive red and black regions by RT-PCR. The same strain used for de novo genome sequencing (F2-5 strain, $h^C$) was used. In total, 12 samples (2 colours [Black/Red] × 2 developmental stages [24 h AP/72 h AP] × 3 biological replicates) were prepared for RNA-seq analysis. Two fragments of bored epidermis from left and right elytra were collected as a single sample in each condition. All total RNA extracted from each sample (12–158 ng) using RNeasy Mini Kit (QIAGEN) and QIAcube (QIAGEN) was used for each cDNA library preparation. RNA-seq library preparation was performed using the SureSelect strand-specific RNA library prep kit (Agilent) according to the manufacturer's instructions. Briefly, mRNA was purified using Oligo-dT Microparticles. The strand-specific RNA-seq libraries were prepared using dUTP and Uracil-DNA-Glycosylase. The libraries and its intermediates were purified and size-fractionated by AMPure XP (Beckman Coulter). For quality check and quantification of the RNA-seq libraries, we employed 2100 Bioanalyzer and DNA 7500 kit (Agilent). 100 bp paired-end read RNA-seq tags were generated using the Hiseq 2500 (Illumina) following the methods in the User Guide.

In advance of reference mapping, adaptor and poly-A sequences were trimmed from raw RNA-seq reads by using Cutadapt (ver. 1.9.1)[67]. Low-quality reads were also filtered out by a custom Perl script as described previously[82]. The preprocessed RNA-seq reads were mapped to the reference *H. axyridis* genome (assembly version 1) using TopHat2[83] (ver. 2.1.0) with default parameters, and assembled by Cufflinks[84] (ver. 2.2.1) with the -u option in each sample. All predicted transcript units and all loci from different samples were merged by Cuffmerge in the Cufflinks suite. The RNA-seq read pairs (fragments) mapped to each predicted transcript unit and locus were counted using HTSeq[85] (ver. 0.6.1) with the options "-s no -t exon -i transcript" and "-s no -t exon -i locus", respectively. The downstream statistical analyses were performed using edgeR[86,87] package (ver. 3.16.5). The raw RNA-seq fragment counts were normalised by the trimmed mean of M-values (TMM) method. Fold change between black and red regions in each stage and its statistical significance (FDR) were calculated. The mean fold changes of the genes in the scaffolds including the $h$ locus candidate region were visualised with IGV[88,89] software (ver. 2.3.88).

**Comparison of the *pannier* locus size**. The holometabolous insects, whose genomic sequences are well assembled at the *pannier* locus to the extent that at least the two paralogous GATA transcription factor genes, *GATAe* and *serpent*, are included in the same scaffold, were selected for comparison.

Concerning Coleoptera, genomic sequences were collected from the Genome database at NCBI[90] (GCA_000002335.3, Tcas5.2; GCA_001937115.1, Atum_1.0; GCA_000390285.2, Agla_2.0; GCA_000648695.2, Otau_2.0; GCA_001412225.1, Nicve_v1.0; GCF_000699045.1, Apla_1.0; GCA_002278615.1, Pchal_1.0) and Fireflybase[91] (*Photinus pyralis* genome 1.3, *Aquatica lateralis* genome 1.3). Concerning holometabolous insect other than Coleoptera, genomic information at Hymenoptera Genome Database[92] (Hymenoptera) (GCF_000002195.4, Amel_4.5; GCF_000217595.1, Lhum_UMD_V04; GCF_000002325.3, Nvit_2.1), Lepbase[93] (Lepidoptera, butterfly; Danaus_plexippus_v3_scaffolds), SilkBase[94] (Lepidoptera, silk moth; Genome assembly (Jan. 2017)), Flybase[95] (Diptera, *Drosophila*, dmel_r6.12_FB2016_04), and the Genome database at NCBI[90] (Diptera, mosquitos; GCA_000005575.1, AgamP3; GCA_002204515.1, AaegL5.0) (Lepidoptera, moth; GCA_002192655.1, ASM219265v1) were utilised. We performed BLAST[72] search (TBLASTN, ver. 2.2.26) using the amino acid sequence of *H. axyridis* Pannier as a query, and identified *pannier* orthologues by focusing on the top hits, and the conserved synteny of the three paralogous GATA transcription factor genes, in which *serpent*, *GATAe*, and *pannier* are tandemly aligned in this order from the 5′ to 3′ direction. The sizes of the upper noncoding region of *pannier* were estimated by calculating the difference between the coordinates of the 3′ end of BLAST hit region of *GATAe* and the 5′ hit region of *pannier*. Traces of translocation or insertions between the paralogous GATA genes were surveyed by looking into the annotations between the *GATAe* and *pannier* loci. If such a genomic rearrangement was found, we recalculated the size of the upstream region of *pannier* using the neighbouring gene. If the exon 1 of *pannier*

was being annotated as an exon distinct from that including the initiation codon as in *H. axyridis*, the size of the first intron was calculated as well.

**Motif enrichment analysis**. To search for the DNA-binding sites of known transcription factors at *pannier*, 1139 DNA-binding motifs of *Drosophila* transcription factors were retrieved from the JASPAR[96] database using the MotifDb[97] R package. Concerning the SD-binding motif, the position weight matrix (PWM) scores were calculated using the 2557 ChIP-seq peaks in *Drosophila* genome obtained by Ikmi et al[98]. and the RSAT peak-motifs program[99]. The nucleotide sequences of the three upper noncoding regions of *pannier* (the upper intergenic region, the upstream half of the first intron, and the downstream half of the first intron) were collected by forging BSgenome[100] data packages using our *H. axyridis* and *C. septempunctata* genome sequences, and by retrieving the sequences using coordinate information obtained for annotation in Fig. 5a and the GenomicFeatures[101] R package. We here defined the upstream half of the first intron as the region including all of the traces of inversions or corresponding sequences shared among different $h$ alleles in *H. axyridis*. GRange objects were generated using the coordinate information obtained for annotation in Fig. 5a. Motif enrichment was quantified using the PWMEnrich[102] R package. As control background genomic regions for the upper intergenic regions of *pannier*, we used 2 kb promoter sequences of 11,279 genes to which RNA-seq reads were mapped (*H. axyridis* genome assembly version 1), but which was not located within the 10 kb from both ends of each genomic scaffold. As control background genomic regions for the upstream and downstream regions of the first intron of *pannier*, we used 2 kb sequences at the 5′ and 3′ end of the first introns, which are more than 2 kb long, and without gaps (2825 and 2810 sequences, respectively). Since there is no reliable gene annotation data for the *C. septempunctata* genome, we used the *H. axyridis* genomic background sequences. Each motif enrichment score, which is related to average time that transcription factors spend in binding to a DNA sequence[103], was calculated using default parameter of PMWEnrichment. *br* and *da* DNA-binding motifs were excluded from the analysis.

**Molecular phylogenetic analysis**. Concerning the coding region of *pannier*, nucleotide sequences of the cloned cDNAs were aligned, and trimmed in the same way (Supplementary Data 5a, b). Concerning the conserved regions in the upper half of the first intron of *pannier*, nucleotide sequences were collected from the BLAST hits obtained to construct Fig. 5a. The collected BLAST hit sequences were arranged in the same directions using a custom Perl script, and aligned using MAFFT[104] (ver. 7.222). We concatenated three alignment blocks, and manually excluded GAP sites and seemingly nonhomologous sites in the alignment (Supplementary Data 5c, d).

The ML phylogenetic trees were constructed using RAxML[105] (ver. 8.0.0) with the options "--maxiterate 1000 --localpair --clustalout". We determined appropriate models of sequence evolution under the AICc4 criterion using Kakusan4[106]. One hundred replicates of shotgun search for the likelihood ratchet were performed. Nodal support was calculated by bootstrap analyses with 1000 replications.

## Data availability

The cloned cDNA sequences were deposited in the DNA Data Bank of Japan (DDBJ) under the accession numbers LC269047–LC269055 (Ha-pnr), LC269056 (Cs-pnr), and LC269057 (Cs-rp49). The genomic sequencing data, the genomic resequencing data (short reads, RAD-seq, linked-reads, and long reads), and the RNA-seq data were deposited in DDBJ under the accession numbers DRA002559, DRA006068, DRA007003, DRA007002, DRA007004, and DRA005777, respectively. The assembled genomic sequences were deposited in DDBJ under the accession numbers BHEG02000001 −BHEG02018515 (*H. axyridis* genome assembly version 1), BHEG02000001 −BHEG02018515 (*H. axyridis* genome assembly version 2), BHEF01000001 −BHEF01044316 (*H. axyridis* linked-read genome assembly, $h^C$, NT6 strain), BHEE01000001–BHEE01050762 (*H. axyridis* linked-read genome assembly, $h^A$, F2 hybrid), BHED01000001–BHED01050274 (*H. axyridis* linked-read genome assembly, $h$, NT8 strain), BHEC01000001–BHEC01055573 (*C. septempunctata* linked-read genome assembly, MD8 strain), and AP018896–AP018898 (the *H. axyridis* gap-filled genomic scaffolds including the *pannier* locus).

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

## Acknowledgements

We thank G. Eguchi, T. Ohde, H. Gotoh, Y. Sato, T. Yaginuma, M. Kobayashi, M. Ikeda for discussions, D.J. Emlen for critical reading of the manuscript, J. Morita, T. Mizutani for experimental support, H. Kawaguchi for rearing of ladybird beetles, H. Asao and A. Akita for library preparation and machine operation of the resequencing analyses, and Functional Genomics Facility, NIBB Core Research Facilities for technical support. Computations were partially performed on the supercomputers at the Data Integration and Analysis Facility, National Institute for Basic Biology and at the ROIS National Institute of Genetics. This study was supported by a Grant-in-Aid from Formation and Recognition, Precursory Research for Embryonic Science and Technology (PRESTO), Japan Science and Technology Agency (JST), MEXT KAKENHI Grant Numbers 18017012, 20017014, 26113708, 221S0002, 17H05848, 18H04828, JSPS KAKENHI Grant Number 22380035, and NIBB Collaborative Research Programs (18-433).

## Author contributions

T.N. and T.A. conceived and designed this study. T.M. and T.N. analysed the elytral pigmentation processes. T.A., T.M., K.G., K.H., A.I. and J.Y. analysed the sequence data. T.A, T.M., K.G., A.I., K.H. and J.H. performed cloning of the *pannier* genes from different alleles and species of ladybirds. T.M., K.G., A.I., K.H. and J.H. performed the larval RNAi experiments. K.H., K.G. and J.H. performed the semi-quantitative RT-PCR. T.A. performed the in situ hybridisation. J.Y. collected the total RNA for the RNA-seq analysis, and the genomic DNA samples for the initial de novo genome assembly. T.A. collected the DNA samples for the resequencing analyses. M.S. and Y.S. collected the RNA-seq raw data. Y.M. and A.T. performed the initial de novo genome assembly. K.Y. and S.S. collected the raw data for the resequencing and the RAD-seq analyses. K.Y. constructed the flexible ddRAD-seq protocol. R.K., M.O. and T.I. performed reassembly of the genome, de novo assembly of the linked-read genomic data, and validation of the obtained genomic scaffolds. M.K., T.T., T.A. and K.Y. performed mapping and quantification of the RNA-seq data. T.A. performed the data analyses for the genetic association studies, the gene annotation, the motif enrichment analysis, and the molecular phylogenetic analyses around the *pannier* locus. T.A. and T.N. wrote, and all authors commented on the manuscript.

## Additional information

**Competing interests:** The authors declare no competing interests.

