## [Peer Review File · Nature Communications]

Reviewers' comments:

Reviewer #1 (Remarks to the Author):

The manuscript "pannier determines highly diverse intraspecific elytral colour patterns of ladybird beetles" provides the long-awaited identification of a gene driving spotted pattern variation in ladybugs. This is potentially a foundational paper that will kickstart a series of studies linking genetics, development and ecology in this system

First the authors provide descriptive data of great interest about color development in the ladybug elytrae. Then, they present mapping data that is more than adequate to support the finding that allelic variation at the pannier explains the phenotypic shifts between the h, hA, and hC alleles. I found awkward, however, that the RNAi data was presented before the mapping, as usually, a forward genetics approach first identifies a candidate gene that is then assessed by reverse genetics, such as a knockdown. If this is really reflective of the experimental chronology that occurred, I would not mind keeping the current order, but typically, Fig. 4 should actually come as one of the first two figures before subsequent developmental genetics work, and loss-of-function data as a final climax?

The combination of mapping, RNA-seq, in situ, and RNAi strongly imply a role for pannier. I have three suggestions for improvement

- ALLELIC SERIES:

the authors say "The elytral colour patterns are formed through the superposition of combinations of two of the four major allelic and dozens of minor allelic colour patterns (more than 20 different allelic patterns in total)"

This is based on literature that dates from before 1950 which I tried to check, and honestly, I am uncomfortable accepting this as evidence that the pannier locus is hosting the variation responsible for 20 patterns in total. Many of those could be due to modifier loci, and without genotyping assays in this older literature, the claims made about the supergene architecture are a rather fuzzy. In fact, page 182 of Komai et al. (ref. 3) indicates tight but incomplete linkage between alleles.

Can the author clarify what is the evidence that pannier drives the HSp allele, while this was not included as an allelic parent in their crosses?

And is it necessary to speculate here that pannier drives more minor alleles, while the mechanism for "mosaic dominance" is clearly far from being understood?

I recommend the authors to streamline their abstract, introduction and discussion to reflect the fact that pannier maps to three alleles for now (h, hC, and hA ; see Methods) and that further work is required to test the hypothesis this locus is a hotspot of phenotypic variation hosting more alleles (Martin and Orgogozo, *Evolution* 2013). For comparison, we initially mapped WntA in three morphs of *Heliconius erato*, 2 morphs of *Heliconius melpomene*, and 2 morphs of *Heliconius cydno* (Martin et al. *PNAS* 2013), and this is later that this gene was found to be the locus of more alleles using Linkage mapping and GWAS (Gallant et al *Nat.*

Comm. 2014, Huber et al. Heredity 2015, Van Belleghem et al. Nat Eco Evo 2017). I trust pannier will map to more alleles, but I do not think it is timely to make that claim yet.

- GENOME SCAFFOLDING AROUND PANNIER:

a big weakness of the paper is due to the lack of scaffold contiguity around pannier (for instance in Fig. 4). For now, the authors can simply not substantiate their claim (here taken from the Abstract) that their data "reveal a ladybird-specific large 150 kb-scale intronic expansion in the pannier locus as a characteristic genomic structure that may have facilitated expansion of elytral colour pattern variation". Perhaps there is indeed something interesting in this first intron, but we want to know what it is instead of beating around the bush. This thus seems to me like a key aspect of the paper that needs to be addressed.

The authors could to deploy alternative sequencing strategies to obtain contiguity across their genetic interval. A possibility would be to isolate High Molecular Weight Genomic DNA from their three strains, and run outsource library preparation and low coverage sequencing using PacBio SMRT cells. The long reads could then be used to scaffold their previous HiSeq short reads. A number of alternatives based on optical mapping or cross-linking exist (10X Genomics, Dovetail Genomics, NanoBioGenomics, fosmid libraries). Ultimately, it would be ideal to obtain the pannier genomic scaffold for the 3 mapped alleles and assess the possible role of structural variation (large indels, short inversions, TEs) in relationship to the phenotype.

- GENETIC INTERVAL

The figure 4 provides insufficient resolution on the genetic interval while the authors have more DNA and markers to genotype.

Could the 140 individuals ($hC = 70$, $hC/hA = 70$) that were pooled could be genotyped individually by PCR at DUS, pnr, and mus201? this could triple their resolution...

Could the authors genotype an additional marker between pnr and mus201? In fact, if intron 1 is very large, it might possible to prove that the first non-coding exon is outside of the interval.

This is an exciting manuscript that deserves publication in Nat. Comm. , and the finding of pannier is well supported. I believe it deserves more efforts on the narrowing of the causal variation around pannier, and perhaps more efforts testing if obvious structural variations may underlie the three alleles.

Signed Review.

Arnaud Martin, Assistant Professor

Reviewer #2 (Remarks to the Author):

Understanding the biology of intraspecific phenotypic diversity has been of considerable interest, from both ecological and genetic perspectives. *Harmonia axyridis* shows incredible phenotypic diversity, which is determined genetically, but is also influenced by environmental temperature.

Ando et al. have made a huge leap forward in understanding how *H. axyridis* can maintain such an array of diversity. Here, they have compared tissue structures during elytral development to show where colours are laid down, knocked out a candidate gene with RNAi (pannier), found differential expression of pannier in different wing sections, sequenced the genome and performed fine-scale linkage analysis. Together this comprehensive analysis shows pannier is required for melanisation and the pannier locus maps precisely to the colour patterning region of the genome. Pannier also determines colour pattern in other ladybirds (*Coccinella septempunctata*) demonstrating functional conservation throughout their evolution. This analysis is really impressive, work is succinctly and clearly explained and I really enjoyed reading it. I only have very minor suggestions/questions.

1. Line 80-81

From the main text, it's not clear how and why you chose pannier as a RNAi target. To test one candidate gene and find it's the right one seems very lucky. As you write "This result was unexpected because pannier is not essential for wing blade patterning in *Drosophila*". If there were other genes tested, perhaps they could be mentioned or put in a supplementary file. Otherwise, the RNAi experiments should follow after other evidence has been presented (genetic mapping).

2. Resequencing of two forms was done; hC (F6 strain) and hA (NT3 strain) but there didn't seem to be any comparative genomics across the pannier gene. It would be nice to know how different these forms are and whether there is any variation in Scalloped binding site number. Considering protein coding changes in pannier don't account for phenotypic diversity, it would have been nice to see a little more on the regulatory regions.

3. Line 31.

It seems a bit unusual stating that previous genetic work was done by 'Asian' geneticists, yet you didn't provide the ethnic background of Theodosius Donzhansky in the previous sentence. Consequently, I would recommend removing the ethnic associations unless they serve a clear scientific purpose.

4. Line 150-151

Claiming Scalloped binding motifs accumulated *H. axyridis* pannier intron 1, relative to other insects, seems like an unfair comparison. Given intron 1 of *Ha* is so much larger than the other insect species investigated, the expectation would be that all binding motifs to have 'accumulated'. A fairer comparison might be to compare the number of scalloped binding sites in the *H. axyridis* pannier 173 kb intron 1 and compare it to the genome average across 173 kb windows. Is there any reason RNAseq expression data was only assessed for

vestigial and not scalloped (Supp. Fig. 7)? Did Scalloped show differential expression?

Michie et al. (2010, DOI: 10.1111/j.1420-9101.2010.02043.x) found that temperature greatly affected the number of spots and melanisation in *H. axyridis succinea* (h/h). In the future, it would be very interesting to see whether pannier expression changes in response to temperature.

Simon Baxter

Reviewer #3 (Remarks to the Author):

This is a review for the manuscript titled "pannier determines highly diverse intraspecific elytral colour patterns of ladybird beetles" submitted to Nature Communications. This addresses the overarching evo-devo question of how genetic variation can shape natural phenotypic variation. To weigh in on this question the authors leverage a well-suited model the ladybird beetle species *Harmonia axyridis* (*H. axy.*). It has been found for *H. axy.* that it has 20 alleles at a single genetic locus responsible for around 200 aposematic elytral colour patterns. The authors use a suite of genetic techniques to demonstrate that this genetic locus is for the conserved GATA transcription factor gene known as pannier. Pannier promotes the formation of black colour while simultaneously suppressing the red colour's formation. Moreover the work implicates the cis-regulatory region of pannier as harboring one of the major genetic variants contributing to the phenotypic variation. Furthermore, the authors extend beyond the intraspecific variation for *H. axy.* and show that pannier seems to have a more ancient role in elytral colour patterning as it seems to similarly shape the colour pattern of the seven spotted ladybird beetle species *Coccinella septempunctata*.

Overall, I found this manuscript to possess a compelling set of results on a compelling and central evo-devo question. While the figures look amazing, their descriptions were generally too vague. Most disappointingly, though, was the absence of clear justified evolutionary model for the inclusion of pannier in elytra colour patterning. I suggest rejecting the manuscript, but I encourage a resubmission with an improved discussion and figure legends as described below.

Major Concerns:

1. The pannier gene in situ presented in Figure 3b is not the most convincing result. Specifically the more intense pattern of expression in regions that develop to be black coloured is not obvious. If a better result cannot be provided, I suggest softening the language on page 13 line 98 to read "pannier seemingly showed higher".
2. The data supporting the co-option of Vestigial/Scalloped is very very speculative based upon some motifs in the non-coding region of pannier. The presentation of this model should be more indicative of its weakly supported nature. Perhaps call this "One of many plausible models". Within this model, when about did pannier evolve to regulate the genes for black and red pigments? This model needs to be elaborated on. It would be beneficial to have a conceptual figure for this model. The co-option of the wing selectors and pannier regulation of red and black genes should date back before the common ancestor shared

with *Coccinella*. Does the RNAi phenotype for *pannier* where black colour is missing from the head suggest *pannier* regulated pigmentation before the co-option event of SD/Vg? This evolutionary model needs to be fleshed out. The discussion lacks clarity and focus on pages 20-21.

3. Page 19. The authors show that the regulatory region of an allele of *pannier* has expanded in *H. axy*. However we know little of what the gene structure is for *Coccinella*. It is likely to be under a conserved regulatory hierarchy. Does it have SD/Vg binding motifs too?

4. The figure legends lack sufficient detail to appreciate what is being shown and its importance. In particular Figure 4a.

Minor Concerns:

Page 5 Lines 29 and 30 are highly redundant with Page 4 Lines 24-26.

Page 5 Line 31, replace "by Asian geneticists" with just "by geneticists"

Page 5 Line 38 and 39, This sentence seemed clunky and do not agree with "it had to be a single gene". I suggest changing it to something like "By elucidating the mechanisms responsible for how this single genetic locus evolved to shape such a strikingly diverse intraspecific colour polymorphism would provide a case-study that bears upon a major evolutionary-developmental biology question; how does morphology evolve?"

Page 11. It is not obvious to how the authors came to test the *Drosophila notum* patterning *pannier* gene.

Page 17 line 141. Change "gene body size of *pannier*" to "size of the *pannier* locus"

Page 17 line 145. Change to "motif of the insect wing"

Page 17 line 146. Can you add a calculation to support the "more accumulated"? Perhaps motifs per 1 kilo base pair.

Page 20 line 179. Sentence does not work. Change to "regulate the multiple intraspecific wing colour patterns."

Page 20 line 180. Change "pathways" to "mechanisms" and change "represent" to "stem from". This is also a run-on sentence that should be chopped into two sentences.

Page 30 line 277. Change "raise" to "increase" and change "penetrance to" to "penetrance in"

Page 31 line 293. You should share the sense probe images to letters readers compare and contrast with the antisense probe signal.

Page 36 line 351. Change "Totally 12" to "In total, 12"

Responses to Reviewers:

First of all, we performed additional sets of *de novo* genome assemblies of h^C , h^A and h alleles in *H. axyridis* and a strain in *C. septempunctata*, and could obtain a contiguous genomic scaffold including *pannier* in each sample. As a result, we found traces of repeated inversions at the 1st intron of *pannier* in *H. axyridis*, which seemed to be the major driving force to have facilitated high diversification of elytral colour patterns in the ancestral lineage of *H. axyridis*. According to this finding, we changed the title of our paper as “Repeated inversions at the *pannier* intron drive diversification of intraspecific colour patterns of ladybird beetle”. Please see our responses to each Reviewer’s comments below, and also please evaluate descriptions on the new findings in the revised version of our manuscript.

Responses to Reviewer #1:

Comment #1

- I found awkward, however, that the RNAi data was presented before the mapping, as usually, a forward genetics approach first identifies a candidate gene that is then assessed by reverse genetics, such as a knockdown. If this is really reflective of the experimental chronology that occurred, I would not mind keeping the current order, but typically, Fig. 4 should actually come as one of the first two figures before subsequent developmental genetics work, and loss-of-function data as a final climax?

The presentation order that Reviewer #1 suggested is actually normal when we select a forward genetic approach. However, the order of the data that we presented in this paper is according to the chronology that we actually experienced. Thus, we did not change the presentation order in the revised version of our manuscript. Alternatively, as pointed out by Reviewer #2 and #3, we inserted description on our initial candidate approach. The actual phenotypes in the first small screening were listed in Supplementary Table 1. (Page 11, Lines 82-83)

Comment #2

- The combination of mapping, RNA-seq, in situ, and RNAi strongly imply a role for pannier. I have three suggestions for improvement

We appreciate Reviewer #1's suggestions. One by one responses to the suggestions are listed below.

Comment #3

- ALLELIC SERIES:

the authors say "The elytral colour patterns are formed through the superposition of combinations of two of the four major allelic and dozens of

minor allelic colour patterns (more than 20 different allelic patterns in total)”

This is based on literature that dates from before 1950 which I tried to check, and honestly, I am uncomfortable accepting this as evidence that the pannier locus is hosting the variation responsible for 20 patterns in total. Many of those could be due to modifier loci, and without genotyping assays in this older literature, the claims made about the supergene architecture are a rather fuzzy. In fact, page 182 of Komai et al. (ref. 3) indicates tight but incomplete linkage between alleles.

- Can the author clarify what is the evidence that pannier drives the HSp allele, while this was not included as an allelic parent in their crosses?

- And is it necessary to speculate here that pannier drives more minor alleles, while the mechanism for “mosaic dominance” is clearly far from being understood?

I recommend the authors to streamline their abstract, introduction and discussion to reflect the fact that pannier maps to three alleles for now (h, hC, and hA ; see Methods) and that further work is required to test the hypothesis this locus is a hotspot of phenotypic variation hosting more alleles (Martin and Orgogozo, Evolution 2013). For comparison, we initially mapped WntA in three morphs of *Heliconius erato*, 2 morphs of *Heliconius melpomene*, and 2 morphs of *Heliconius cydno* (Martin et al. PNAS 2013), and this is later that this gene was found to be the locus of more alleles using Linkage mapping and GWAS(Gallant et al Nat. Comm. 2014, Huber et al. Heredity 2015, Van Belleghem et al. Nat Eco Evo 2017). I trust pannier will map to more alleles, but I do not think it is timely to make that claim yet.

In *H. axyridis*, Hosino (1934) first described that the major four alleles are genetically linked to the same genetic locus based on his genetic crossing experiments. Successive genetic experiments by Hosino (Hosino, 1936, 1939, 1940, 1941, 1942, 1943a, 1943b, 1948; Genetic studies of the lady-bird beetle, *Harmonia axyridis* PALLAS. [Report II-IX]) indicated that 20 different elytral patterns are tightly genetically linked to the previously identified 4 loci, implying existence of the corresponding 24 alleles. Tan

(1946) systematically genotyped 2,972 F2 individuals of 15 patterns of elytral colour patterns in total using the most recessive allele *succinea* (*s*) (denoted as *h* allele in our paper) as a testing allele, and concluded that those 15 patterns are regulated by corresponding alleles located at the same genetic locus (*h*; denoted as *s* by Tan). Komai considered that *h* might not correspond to a single gene based on the single rare F2 heterozygotes seemingly possessing three alleles ($S^E/S^R/s$) reported in Tan (1946).

Now, we can revisit investigation of these different alleles focusing on the *pannier* locus, but actually, as pointed by Reviewer #1, it is still not the timely stage to conclude that all colour pattern morphs are regulated by *pannier*. Therefore, we reflected the fact that *pannier* maps to the major four alleles at present (*h*, *h^C*, *h^A* and *h^{Sp}* [deduced from our new data]; see below for the detail) in the abstract, introduction and discussion. We replaced the expression “the single genetic locus” for “the tightly linked genetic locus” (Page 2, Line 6; Page 5, Line 30; Page 6, Line 39; Page 35, Line 332), and change the wordings around the expression.

We also mentioned that further work is required to test the hypothesis that this locus is a hotspot of morphological evolution in ladybird beetles, at the end of the Discussion (Page 35-36, Lines 335-337).

- Can the author clarify what is the evidence that *pannier* drives the HSp allele, while this was not included as an allelic parent in their crosses?

In addition to the individual genotyping of F2 progenies in *h-h^C* F0 cross, we further performed linkage analyses using the *h^A-h^{Sp}* F0 combination (n = 273), and the *h^A-h^C* F0 combination (n = 183) (individual genotyping and RAD-seq, respectively). As a result, we found that all of the four major alleles presented in this paper are tightly linked to the *pannier* locus. We included this result in the revised version of our manuscript. (Page 15, Lines 117-123; Figure 4a; Supplementary Table 2)

Comment #4

- GENOME SCAFFOLDING AROUND PANNIER:

a big weakness of the paper is due to the lack of scaffold contiguity around pannier (for instance in Fig. 4). For now, the authors can simply not substantiate their claim (here taken from the Abstract) that their data “reveal a ladybird-specific large 150 kb-scale intronic expansion in the pannier locus as a characteristic genomic structure that may have facilitated expansion of elytral colour pattern variation”. Perhaps there is indeed something interesting in this first intron, but we want to know what it is instead of beating around the bush. This thus seems to me like a key aspect of the paper that needs to be addressed.

The authors could to deploy alternative sequencing strategies to obtain contiguity across their genetic interval. A possibility would be to isolate High Molecular Weight Genomic DNA from their three strains, and run outsource library preparation and low coverage sequencing using PacBio SMRT cells. The long reads could then be used to scaffold their previous HiSeq short reads. A number of alternatives based on optical mapping or cross-linking exist (10X Genomics, Dovetail Genomics, NanoBioGenomics, fosmid libraries). Ultimately, it would be ideal to obtain the pannier genomic scaffold for the 3 mapped alleles and assess the possible role of structural variation (large indels, short inversions, TEs) in relationship to the phenotype.

We additionally performed linked-read and long read analyses of *h*, *h^A* and *h^C* alleles, and *C. septempunctata* using PacBio (approximate mean coverage: 10x) and 10x Genomics Chromium platforms (approximate mean coverage: 200x [Hiseq X ten]), and obtained contiguous scaffolds including *pannier* and neighboring genes. Mainly, four new findings were obtained from this analysis:

- (1) The first intronic sequences of *pannier* are highly diverged among *h*, *h^A* and *h^C* alleles compared to the neighboring genomic sequences, and contained traces of repeated inversion within the 1st introns (Fig. 5a);
- (2) Molecular phylogenetic analysis using the conserved intronic sequences revealed that the *h* allele and the common ancestor of other three alleles

- diverged first during evolution, and the latter three alleles diverged recently (Fig. 5d);
- (3) Several sequence blocks in the intron are conserved in *C. septempunctata* as well (Fig. 5a, *C. sep*);
 - (4) Repertoires of known DNA binding motifs in the *h*, *h^A* and *h^C pannier* intronic regions are also highly diverged among the alleles (Table 1a).

We included the results described above in our revised manuscript, and modified the discussion accordingly. (Result: Page 17, Lines 149-157; Pages 18-26 166-235; Page 228, 256-266) (Discussion: Pages 29-34, 275-316)

Comment #5

- GENETIC INTERVAL

The figure 4 provides insufficient resolution on the genetic interval while the authors have more DNA and markers to genotype.

Could the 140 individuals (hC = 70, hC/hA = 70) that were pooled could be genotyped individually by PCR at DUS, pnr, and mus201? this could triple their resolution...

Could the authors genotype an additional marker between pnr and mus201? In fact, if intron 1 is very large, it might possible to prove that the first non-coding exon is outside of the interval.

We additionally genotyped the BC1 individuals used in the bulked segregant analysis in the previous version of our manuscript, using RAD-seq (n = 183). This analysis could narrow down the 3' downstream genomic region outside of *pannier*, but the *pannier* locus was still inside the candidate region. Thus, we also additionally genotyped F2 progenies in a novel *h^A · h^{Sp}* F0 cross by PCR-based individual genotyping (n = 273). By assembling the data from the three crosses (*h · h^C*, *h^C · h^A*, *h^{Sp} · h^A*), we succeeded in tripling the mapping resolution (Page 15, Lines 117-123; Figure 4a; Supplementary Table 2). Still, size of the *pannier* locus is within the size of the responsive region that we could narrow down. Instead of linkage analysis, we discuss the possible importance of the *pannier* intronic regions, and neighboring

non-coding DNA by comparing the newly assembled genomic scaffold sequences of h^A , h^C and h alleles obtained by long read and linked-read analyses, as we explained in the previous response to Reviewer #1's Comment #3.

Comment #6

This is an exciting manuscript that deserves publication in Nat. Comm. , and the finding of pannier is well supported. I believe it deserves more efforts on the narrowing of the causal variation around pannier, and perhaps more efforts testing if obvious structural variations may underlie the three alleles.

Thank you very much for the comment. We additionally performed a RAD-seq analysis, and individual genotyping using three sets of genetic crosses in total to narrow down the responsive genomic regions. Furthermore, we performed *de novo* genomic assembly of h^C , h^A , and h alleles in *H. axyridis*, and *C. septempunctata* to reveal structural variations around the *pannier* locus. We believe that the quality of our manuscript is now much improved thanks to your comments.

Responses to Reviewer #2:

- I only have very minor suggestions/questions.

Comment #1

- 1. Line 80-81

From the main text, it's not clear how and why you chose pannier as a RNAi target. To test one candidate gene and find it's the right one seems very lucky. As you write "This result was unexpected because pannier is not essential for wing blade patterning in Drosophila". If there were other genes tested, perhaps they could be mentioned or put in a supplementary file. Otherwise, the RNAi experiments should follow after other evidence has been presented (genetic mapping).

We initially screened 10 genes related to wing/body wall patterning (*pannier*, *decapentaplegic*, *wingless*, *Cubitus interruptus*, *apterous*, *Distal-less*, *aristaless*, *blistered*, *araucan*, *Epidermal growth factor receptor*), and found that *pannier* affected colour patterns the most drastically. We summarised qualitative descriptions of phenotypes in this small screening in Supplementary Table 1, and mentioned this screening at (Page 12, Lines 82-83).

Comment #2

- 2. Resequencing of two forms was done; hC (F6 strain) and hA (NT3 strain) but there didn't seem to be any comparative genomics across the pannier gene. It would be nice to know how different these forms are and whether there is any variation in Scalloped binding site number. Considering protein coding changes in pannier don't account for phenotypic diversity, it would have been nice to see a little more on the regulatory regions.

As described in the response to Reviewer#1's Comment #4, we additionally sequenced genomic sequences of three different alleles (h^A , h^C

and *h*) in *H. axyridis* and one strain in *C. septempunctata*, and found the major differences at the *pannier* locus. Based on this result, we discussed possible regulatory differences associated with the phenotype. In our previous manuscript, we discussed that enrichment of Sd-binding sites in the *H. axyridis* at the *pannier* locus may be associated with acquisition of novel expression domain in the ladybird beetle's elytra (forewing). However, our new motif-enrichment analysis suggested that Vg-Sd complex may rather function as an *h^C*-specific patterning factor. Please read our revised results and discussion in the manuscript (Pages 22-25, Lines 211-227).

Comment #3

- 3. Line 31.

It seems a bit unusual stating that previous genetic work was done by 'Asian' geneticists, yet you didn't provide the ethnic background of Theodosius Donzhansky in the previous sentence. Consequently, I would recommend removing the ethnic associations unless they serve a clear scientific purpose.

We eliminated the ethnic associations. (Page 5, Line 29)

"Successive genetic analyses⁵⁻⁷ revealed that ..."

Comment #4

- 4. Line 150-151

Claiming Scalloped binding motifs accumulated *H. axyridis* pannier intron 1, relative to other insects, seems like an unfair comparison. Given intron 1 of *Ha* is so much larger than the other insect species investigated, the expectation would be that all binding motifs to have 'accumulated'. A fairer comparison might be to compare the number of scalloped binding sites in the *H. axyridis* pannier 173 kb intron 1 and compare it to the genome average across 173 kb windows. Is there any reason RNAseq expression data was only assessed for vestigial and not scalloped (Supp. Fig. 7)? Did Scalloped show differential expression?

We expanded the analysis window outside of the *pannier* locus (the upstream intergenic region), and also analysed other known DNA binding motifs enriched in those regions. In this analysis, we used 2 kb promoter sequences or 2 kb intronic sequences of the predicted genes except *pannier* in the *H. axyridis* genome as control sequences, and assessed enrichment of specific known DNA binding motifs in the non-coding sequences around the *pannier* locus. We calculated average affinity score of each sequence (which is related to average time that a Transcription Factor spends in binding to a DNA sequence [Stormo, G. (2000). *Bioinformatics*, 16(1):16–23.]) using the EnrichmentMotif R package. Please see details in the Result (Pages 22-26, Lines 210-235; Table 1).

We showed the expression levels of *vg* because this gene is the only transcription factor gene statistically upregulated at 24h AP. Therefore, *sd* was not significantly upregulated at this stage (data not shown). We inserted additional explanation to clarify this point as follows: “Furthermore, the RNA-seq data for the h^C background also revealed that the *sd* co-activator gene *vestigial* was the only transcription factor gene that was significantly upregulated in the future black region from early pupal stages (Supplementary Figure 6)” (Page 25, Lines 222-224).

Comment #5

- Michie et al. (2010, DOI: 10.1111/j.1420-9101.2010.02043.x) found that temperature greatly affected the number of spots and melanisation in *H. axyridis succinea* (h/h). In the future, it would be very interesting to see whether *pannier* expression changes in response to temperature.

Thank you very much for the comment on plasticity of elytral colour patterns in response to temperature. Temperature-dependent plasticity of the size of each black spot in *h* allele is also reported in Hosino (1942), *Jap. Jour. Genet.* 18: 285-296. We also have been thinking this phenomenon will be very intriguing if we could investigate dynamics of *pannier* expression at

different

temperature.

Responses to Reviewer #3:

- Overall, I found this manuscript to possess a compelling set of results on a compelling and central evo-devo question. While the figures look amazing, their descriptions were generally too vague. Most disappointingly, though, was the absence of clear justified evolutionary model for the inclusion of pannier in elytra colour patterning. I suggest rejecting the manuscript, but I encourage a resubmission with an improved discussion and figure legends as described below.

- Major Concerns:

Comment #1

- 1. The pannier gene in situ presented in Figure 3b is not the most convincing result. Specifically the more intense pattern of expression in regions that develop to be black coloured is not obvious. If a better result cannot be provided, I suggest softening the language on page 13 line 98 to read “pannier seemingly showed higher”.

According to the suggestion, we softened the language in the description of data on the *in situ* expression of *pannier*. (Page 14, Line 106)

Comment #2

- 2. The data supporting the co-option of Vestigial/Scalloped is very very speculative based upon some motifs in the non-coding region of pannier. The presentation of this model should be more indicative of its weakly supported nature. Perhaps call this “One of many plausible models”. Within this model, when about did pannier evolve to regulate the genes for black and red pigments? This model needs to be elaborated on. It would be beneficial to have a conceptual figure for this model. The co-option of the wing selectors and pannier regulation of red and black genes should date back before the common ancestor shared with Coccinella. Does the RNAi phenotype for

pannier where black colour is missing from the head suggest pannier regulated pigmentation before the co-option event of SD/Vg? This evolutionary model needs to be fleshed out. The discussion lacks clarity and focus on pages 20-21.

According to the suggestion, we summarised the possible evolutionary model underlying diversification of the intraspecific wing colour patterns of extant ladybird beetles in the Discussion (Pages, 29-34), and made a figure explaining it (Figure 6). In our revised discussion, we included the results obtained from our additional *de novo* genome assemblies of h^A , h^C , h^{Sp} alleles in *H. axyridis*, and a strain in *C. septempunctata*. The major new findings are as follows (We here repeat our previous response to Reviewer#1's Comment #4):

- “(1) The first intronic sequences of *pannier* are highly diverged among h , h^A and h^C alleles compared to the neighboring genomic sequences, and contained traces of repeated inversion within the introns. (Fig. 5a);
- (2) Molecular phylogenetic analysis using the conserved intronic sequences revealed that the h allele and the common ancestor of other three alleles diverged first during evolution, and other three alleles diverged recently;
- (3) Several intronic sequences are conserved in *C. septempunctata* as well;
- (4) Repertoires of known DNA binding motifs in the h , h^A and h^C *pannier* intronic regions are also highly diverged among the alleles.”

Based on our new genomic data, we inferred the order of emergence among the 4 alleles in *H. axyridis*. In addition, we estimated that repeated inversion events within the 1st intron of *pannier* can be the major driving force to generate and maintain diverse colour patterns within a species. Please read our revised discussion (Pages, 29-34).

The "head" phenotype mentioned in the comment is to be exact the "prothoracic" phenotype. Anyway, we think that this prothoracic phenotype is a key trait to estimate the origin of the colour-patterning function of *pannier*. However, in order to address this issue, we need to examine

prothoracic colour-patterning function in other ladybird beetles, because we do not know whether the prothoracic colour patterning function observed only in *H. axyridis* corresponds to the ancestral function or a derived function at present. We think that this issue should be addressed in future research, but not in this paper.

Comment #3

- 3. Page 19. The authors show that the regulatory region of an allele of pannier has expanded in H. axy. However we know little of what the gene structure is for Coccinella. It is likely to be under a conserved regulatory hierarchy. Does it have SD/Vg binding motifs too?

We tried reconstructing *Coccinella pannier* locus by additional linked-read sequencing analysis, and obtained a scaffold including the entire *pannier* locus. Unlike our previous discussion, enrichment of SD binding motif seemed to be h^C allele-specific character of the *pannier* intron. Alternatively, we surveyed the known *Drosophila* DNA-binding motifs comprehensively, and included the result in our revised manuscript. Please see details in the manuscript. (Pages 22-26, Lines 203-234; Page 27, 244-254)

Comment #4

- 4. The figure legends lack sufficient detail to appreciate what is being shown and its importance. In particular Figure 4a.

We additionally wrote down the detailed explanation of panels in the figure legend for Figure 4a. (Page 16)

- Minor Concerns:

Comment #5

- Page 5 Lines 29 and 30 are highly redundant with Page 4 Lines 24-26.

We revised the latter explanation to avoid redundant expression (Page 5, Lines 26-29).

Comment #6

- Page 5 Line 31, replace “by Asian geneticists” with just “by geneticists”

We eliminated the ethnic association, as also pointed out by Reviewer #2 (Page 5, Line 29).

Comment #7

- Page 5 Line 38 and 39, This sentence seemed clunky and do not agree with “it had to be a single gene”. I suggest changing it to something like “By elucidating the mechanisms responsible for how this single genetic locus evolved to shape such a strikingly diverse intraspecific colour polymorphism would provide a case-study that bears upon a major evolutionary-developmental biology question; how does morphology evolve?”

We appreciate the comment on the sentence concerning our main scope in this study. The suggested expression is consistent with our research scope, and appears to us to be better than our previous sentence. The previous sentence was replaced with the suggested one with some modifications: “Elucidating the DNA structure and the mechanisms underlying the evolution of this tightly linked genetic locus that encodes such a strikingly diverse intraspecific colour pattern polymorphism would provide a case-study that bears upon a major evolutionary-developmental biology question; how does morphology evolve?” (Page 6, Lines 39-42).

Comment #8

- Page 11. It is not obvious to how the authors came to test the *Drosophila notum* patterning *pannier* gene.

As mentioned in the response to Reviewer #2, we screened 10 genes related to wing/body wall patterning genes (*pannier*, *decapentaplegic*, *wingless*, *Cubitus interruptus*, *apterous*, *Distal-less*, *aristaless*, *blistered*, *araucan*, *Epidermal growth factor receptor*). We included qualitative descriptions of phenotypes in this small screening in Supplementary Table 1 (Page 11, Lines 82-83).

Comment #9

- Page 17 line 141. Change “gene body size of *pannier*” to “size of the *pannier* locus”

Because we thoroughly revised our results and discussion about the size of the *pannier* locus, this sentence was deleted from the manuscript.

Comment #10

- Page 17 line 145. Change to “motif of the insect wing”

We changed the wording as suggested (Page 23, Line 214).

Comment #11

- Page 17 line 146. Can you add a calculation to support the “more accumulated”? Perhaps motifs per 1 kilo base pair.

In our revised manuscript, we calculated affinity scores of not only Scalloped but also of the other known transcription factors (which is related to average time that a transcription factor spends in binding to a DNA

sequence [Stormo, G. (2000). *Bioinformatics*, 16(1):16–23.] using the EnrichmentPWM R package. In addition, as mentioned in the response to Reviewer #2's Comment #4, we expanded the analysis window outside of the *pannier* locus (the upstream intergenic region).

(The below sentences are a repeat of the response to Reviewer2's Comment # 4)

“In this analysis, we used 2 kb promoter sequences or 2 kb intronic sequences of the predicted genes except *pannier* in the *H. axyridis* genome as control sequences, and assessed enrichment of specific known DNA binding motifs in the noncoding sequences around the *pannier* locus. We calculated average affinity score of each sequence (which is related to average time that a transcription factor spends in binding to a DNA sequence [Stormo, G. (2000). *Bioinformatics*, 16(1):16–23.] using the EnrichmentMotif R package.”

The results are described in Pages 22-26 & 27 (Lines 210-235 & 245-255), as mentioned in the above response to Reviewer #3's Comment #3.

Comment #12

- Page 20 line 179. Sentence does not work. Change to “regulate the multiple intraspecific wing colour patterns.”

We changed the wording as suggested (Page 34, Lines 321-322).

Comment #13

- Page 20 line 180. Change “pathways” to “mechanisms” and change “represent” to “stem from”. This is also a run-on sentence that should be chopped into two sentences.

We changed the wordings as suggested, and split the sentence into two sentences. (Pages 34-35, Lines 322-324)

Comment #14

- Page 30 line 277. Change “raise” to “increase” and change “penetrance to” to “penetrance in”

We changed the wording as suggested (Page 44, Lines 434).

Comment #15

- Page 31 line 293. You should share the sense probe images to letters readers compare and contrast with the antisense probe signal.

We inserted sense probes images in Figure 3b (Page 13).

Comment #16

- Page 36 line 351. Change “Totally 12” to “In total, 12”

We changed the wording as suggested. (Page 68, Line 695)

Additional comments for Editor and Reviewers:

Although the genomic assembler, Platanus2, is not open currently, we plan to release the same version (2.0.0) of Platanus2 at the web site of Platanus (<http://platanus.bio.titech.ac.jp/>) by the publication of this study.

The additional genomic raw data obtained in this study are now under curation at DNA Databank of Japan (DDBJ). We will include the accession number for the data in the manuscript by the publication of this study. Also we are planning to release all genome assemblies obtained in this study at a website.

REVIEWERS' COMMENTS:

Reviewer #1 (Remarks to the Author):

I have impressed by this extensive revision of the manuscript by Ando et al. The authors addressed all my criticism, streamlining their claims about poly-allelism, improving the genetic resolution of their linkage mapping, and adding genomic sequencing efforts (including long read data) that led to an improved contiguity of the pannier locus, and to the discovery of interesting and multiple intronic inversions, associated with the phenotypes.

They also added an outgroup (*Coccinella*), which allowed them to give some insights on the evolutionary history of the locus. While the model presented in Fig. 6 is not perfect, but it is testable, correctly discussed, and I am largely in favor of keeping it as it is without further data.

Reviewer 3 commented on the quality of the in situ analysis, and I must add I found the data actually quite good given that this elytral tissue is too sclerotized to allow proper penetration of reagents.

I support Publication of the revised manuscript, with the recommendation that the authors add a quick dot-plot analysis of the pannier first intron sequence comparisons to better show the inversions, in their final supplementary data.

Signed Review,
Arnaud Martin

Reviewer #2 (Remarks to the Author):

Here the authors find pannier is up-regulated in black elytral wings and knocking down pannier expression with RNAi removes black, demonstrating a functional link between gene and phenotype. Genetic crosses show the h colour locus is linked with pannier and inversion mutations within the gene are associated with different phenotypes. This is a compelling study and the authors have addressed my previous comments well.

I think the order of experiments is fine, given it is now clear you tested several candidate genes. Providing the subsequent genetic mapping is then appropriate, as this indicates pannier is the colour switch, rather than one of the many genes involved in colour development. Identifying a series of inversions linked with color was a very interesting discovery.

Minor comments.

1. Title may need revision as pannier has several introns
E.g. "Inversions within a pannier intron..."
2. Line 38. Consider deleting 'yet'
3. Line 148 Figure 4c. None of the gene models shown here seem to have exon 1. It would be good to indicate exon 1 and intron 1 for clarity.
4. Line 150. Add the word "performed". "...genome assembler (Platanus2), and performed additional de novo.."
5. Line 182 program rather than programme
6. Figure a is a little confusing to me, as it's not clear if all sequences are being compared to hc(F2-3), or if each sequence is being compared relative to the one below it. It would also be great to have the intron 1 region indicated in this image, or to indicate exonic sequence, which should be conserved.
7. Line 203-206. It would be good to define the upstream sequence more clearly. Do you mean the sequence between GATAe stop codon and pannier ATG start codon?
8. Line 320-321. *Heliconius melpomene* and *H. erato* have multiple, major colour pattern loci. *Heliconius numata* has a single major locus, with minor effects from other loci. *Harmonia* may be somewhat similar to *H. numata*, so perhaps refer to specific *Heliconius* species when discussing the 'contrast' between butterflies and ladybirds.
9. Check for spelling errors

Reviewer #3 (Remarks to the Author):

The critiques of the reviewers were sufficiently addressed in this revised manuscript, and I therefore recommend accepting the revised manuscript for publication.

Responses to Reviewers:

Reviewer #1 (Remarks to the Author):

I have impressed by this extensive revision of the manuscript by Ando et al.

The authors addressed all my criticism, streamlining their claims about poly-allelism, improving the genetic resolution of their linkage mapping, and adding genomic sequencing efforts (including long read data) that led to an improved contiguity of the pannier locus, and to the discovery of interesting and multiple intronic inversions, associated with the phenotypes.

They also added an outgroup (Coccinella), which allowed them to give some insights on the evolutionary history of the locus. While the model presented in Fig. 6 is not perfect, but it is testable, correctly discussed, and I am largely in favor of keeping it as it is without further data.

Reviewer 3 commented on the quality of the in situ analysis, and I must add I found the data actually quite good given that this elytral tissue is too sclerotized to allow proper penetration of reagents.

I support Publication of the revised manuscript, with the recommendation that the authors add a quick dot-plot analysis of the pannier first intron sequence comparisons to better show the inversions, in their final supplementary data.

Thank you very much for the comments. We performed an additional dot-plot analysis around the *pannier* locus (Supplementary Figure 5), and clarified that the traces of the inversions are located within the first intron of the longest *pannier* transcript (*pannier 1A*).

Reviewer #2 (Remarks to the Author):

Here the authors find pannier is up-regulated in black elytral wings and knocking down pannier expression with RNAi removes black, demonstrating a functional link between gene and phenotype. Genetic crosses show the h colour locus is linked with pannier and inversion mutations within the gene are associated with different phenotypes. This is a compelling study and the authors have addressed my previous comments well.

I think the order of experiments is fine, given it is now clear you tested several candidate genes. Providing the subsequent genetic mapping is then appropriate, as this indicates pannier is the colour switch, rather than one of the many genes involved in colour development. Identifying a series of inversions linked with color was a very interesting discovery.

Thank you very much for the comments. Point-by-point responses to each comment are listed below.

Minor comments.

1. Title may need revision as pannier has several introns
E.g. “Inversions within a pannier intron...”

As suggested, we revised the title of our paper, as “Repeated inversions within a *pannier* intron drive diversification of intraspecific colour patterns of ladybird beetles”.

2. Line 38. Consider deleting ‘yet’

We deleted ‘yet’ in the revised manuscript. (Page 9, Line 74)

3. Line 148 Figure 4c. None of the gene models shown here seem to have exon 1. It would be good to indicate exon 1 and intron 1 for clarity.

We inserted exon numbers in the gene model, and made the arrow heads larger (Figure 4c). Also, we inserted a gene model of *pannier* at the top of Figure 5a, and indicated the position of intron 1 with an asterisk.

4. Line 150. Add the word “performed”. “...genome assembler (Platanus2), and performed additional de novo..”

We added “performed” in the sentence. (Page 13, Line 123)

5. Line 182 program rather than programme

We changed the word as suggested. (Page 88, Line 1092)

6. Figure a is a little confusing to me, as it’s not clear if all sequences are being compared to hc(F2-3), or if each sequence is being compared relative to the one below it. It would also be great to have the intron 1 region indicated in this image, or to indicate exonic sequence, which should be conserved.

In Figure 5a, each sequence is compared relative to the one below it. To clarify this point, we modified the sentence explaining it in the figure legend. (Page 88, Line1094-1096). In addition, to show the position of the 1st intron, we inserted a gene model of *pannier* at the top of Figure 5a, and indicated the position of intron 1 by an asterisk.

As suggested by Reviewer #1, we also performed an additional dot-plot analysis around the *pannier* locus, and clarified that the traces of the inversions are located within the first intron of the longest *pannier* transcript. (Supplementary Figure 5)

7. Line 203-206. It would be good to define the upstream sequence more clearly. Do you mean the sequence between GATAe stop codon and pannier ATG start codon?

We defined the upstream sequence as the sequence between the 5' end of 5'UTR of *pannier* and the 3' end of 3'UTR of *GATAe*. We clarified this point in the sentence. (Page 17, Lines 162-163)

8. Line 320-321. Heliconius melpomene and H. erato have multiple, major colour pattern loci. Heliconius numata has a single major locus, with minor effects from other loci. Harmonia may be somewhat similar to H. numata, so perhaps refer to specific

Heliconius species when discussing the ‘contrast’ between butterflies and ladybirds.

We assumed *Heliconius melpomene* and *H. erato* as representative species of *Heliconius* butterflies in the previous manuscript. We clarified this point in the revised manuscript. (Page 27, Lines 271-274)

9. Check for spelling errors

We again checked spelling errors carefully.

Reviewer #3 (Remarks to the Author):

The critiques of the reviewers were sufficiently addressed in this revised manuscript, and I therefore recommend accepting the revised manuscript for publication.

Thank you very much for the comment. We appreciate that our manuscript was much improved thanks to the reviewers' comments.